# FeLMi : Few shot Learning with hard Mixup

**Aniket Roy**
Johns Hopkins University
Baltimore, USA
aroy28@jhu.edu

**Anshul Shah**
Johns Hopkins University
Baltimore, USA
ashah95@jhu.edu

**Ketul Shah**
Johns Hopkins University
Baltimore, USA
kshah33@jhu.edu

**Prithviraj Dhar**
Reality Labs, Meta
Sunnyvale, USA
prithvirj95@meta.com

**Anoop Cherian**
MERL
Cambridge, MA
cherian@merl.com

**Rama Chellappa**
Johns Hopkins University
Baltimore, USA
rchella4@jhu.edu

## Abstract

Learning from a few examples is a challenging computer vision task. Traditionally, meta-learning-based methods have shown promise towards solving this problem. Recent approaches show benefits by learning a feature extractor on the abundant base examples and transferring these to the fewer novel examples. However, the finetuning stage is often prone to overfitting due to the small size of the novel dataset. To this end, we propose **Fe**w shot **L**earning with hard **Mi**xup (**FeLMi**) using manifold mixup to synthetically generate samples that helps in mitigating the data scarcity issue. Different from a naïve mixup, our approach selects the hard mixup samples using an uncertainty-based criteria. To the best of our knowledge, we are the first to use hard-mixup for the few-shot learning problem. Our approach allows better use of the pseudo-labeled base examples through base-novel mixup and entropy-based filtering. We evaluate our approach on several common few-shot benchmarks - FC-100, CIFAR-FS, miniImageNet and tieredImageNet and obtain improvements in both 1-shot and 5-shot settings. Additionally, we experimented on the cross-domain few-shot setting (miniImageNet → CUB) and obtain significant improvements. Code: https://github.com/aniket004/Felmi

## 1 Introduction

Supervised deep learning has revolutionized data-driven computer vision tasks, e.g., object detection, segmentation [5], etc. However, it is expensive to collect and annotate large amounts of data, especially in some fields like medical image diagnosis where one needs to rely on experts for annotating the data. Despite recent progress in learning from large data, learning robust models using a few labeled examples still remains quite challenging. Few-shot learning (FSL) [31, 28, 33] deals with this problem, where there exists base classes with plenty of training examples, and novel classes with only a few training samples. The goal is to learn a representation from the base classes, then quickly and efficiently adapt to the novel classes from few examples.

There are two paradigms for training few-shot learner - (1) episodic training and (2) transfer learning. During episodic training, in each episode, a few examples from the training data are sampled to form the "support set", on which training is performed and evaluated on the "query set" drawn from the test dataset. Generally, the model is trained for a large number of episodes to get better generalization. On the one hand, this approach is both time consuming and also incurs inductive (prior knowledge driven bias) and sampling bias. On the other hand, recent approaches use simple transfer learning to the novel classes and obtained comparative performance. Tian et al. [31] show that pretraining on the large scale base dataset to learn a good representation, followed by finetuning on the few

novel class examples using a simple linear classifier significantly outperforms the more complicated meta-learning approaches in most of the FSL benchmarks. The performance is further boosted using self-distillation and distillation across base and novel class representations [21].

Data scarcity is a fundamental issue in few-shot learning. To circumvent this, AssoAlign [1] uses the nearest neighbor base examples in addition to the novel examples with adversarial augmentation. Jian et al. [11] generates pseudo-labels for the base class samples and retrains on the entire dataset by "hallucinating" the labels of the base class samples. However, literature in semi-supervised learning has shown that poorly pseudo-labeled samples degrade the performance [21].

We overcome the above issues in these prior approaches using entropy filtering on the noisy pseudo-labels. This step helps improve the training by discarding noisy pseudo-labeled base samples during training. Next, we propose a solution to the data scarcity issue by synthetically generating training examples with both novel and base examples using a mixup strategy. The mixup samples are generated with minimal overhead while adding the much needed regularization by synthesizing new points close to the novel distribution. Motivated by the efficacy of hard example mining in computer vision [26], we propose to improve the naïve mixup step with uncertainty guided hard-mixup. Further, we propose a technique to make use of base samples for mixup in addition to the novel samples. Note that Mangla et. al [16] has also used mixup for few-shot learning, but the motivation is completely different in our setting. Their approach does not add any more samples to training and use it as a regularizer across layers. On the contrary, we are the first to propose generating hard mixup samples to tackle the data scarcity issue in FSL.

Another contribution of our work is that we redefine hard-mixup for the few-shot learning setup. Note that hard-mixup has been used in contrastive learning, where positive and negative classes are already defined [12]. In contrast, for N-way K-shot problem, where there exists N classes, the notion of hard examples is not straightforward. While, hard mixup similar to these approaches could be extended in our setup by mixing and assigning a hard label, we instead work with soft labels and select the optimal examples to mix with the appropriate lambda values. To this end, we define hard example mixing based on a margin-based uncertainty measure, inspired by the concept of active learning. We use the margin to estimate the hardness of a sample, thereby measuring the uncertainty in predicting a particular class with high probability.

In summary, the contributions of this work are as follows:

1. To handle the data scarcity problem in few-shot learning, we use mixup-based data augmentation.

2. We perform hard mixup using a novel margin-based uncertainty measure, which further elevates performance. To the best of our knowledge, we are the first to use hard-mixup for few-shot learning.

3. We propose an approach (FeLMi) to make use of both base and novel examples for generating more samples for novel training.

4. We validate our approach through extensive experiments on standard few-shot benchmarks - FC100, CIFAR-FS, miniImageNet and tieredImageNet demonstrating state-of-the-art results. Our approach also demonstrates promising improvements on the standard miniImageNet $\rightarrow$ CUB cross domain benchmark.

## 2   Related Work

Few-shot learning methods can be divided into two broad categories, (1) Meta-learning, (2) Transfer learning methods.

**Meta Learning.** Traditionally, meta-learning based methods have been effective for few-shot learning tasks and usually consists of a meta-training and a meta-testing phase. In both meta-training and testing phases, multiple episodes are sampled from a task distribution and the model is trained on the support samples of the episode and evaluated on the query set. Thus, the model is trained on unseen classes via a learning to learn approach. Meta-learning methods can also be subdivided into two categories, viz., metric-based meta learning and optimization-based meta-learning.

Metric-based meta-learning methods predict the label of the query as a weighted sum of the labels over the support samples. Popular metric-based meta-learning methods include Prototypical Networks [28],

Relation Networks [30], Matching networks [33], TADAM [17], etc. Optimization-based meta-learning methods adapt the model parameters using a small number of gradient steps, e.g., MAML [9], LEO [23], etc. MetaOptNet [19] solves a differentiable convex optimization problem for few-shot learning with better generalization. Other meta-learning methods use earth-mover's distance [38], set-to-set functions [35], human interpretable concepts [5], etc.

**Transfer learning methods.** Recent methods have shown that contrastive pretraining on large base example dataset and simple finetuning on the scarce novel examples performs surprisingly better than the complex meta-learning-based methods. As an example, RFS [31] outperforms all the meta learning baselines by learning a self-supervised representation followed by a simple logistic classifier. Following a similar trend, SKD [18] uses rotational self-supervised distillation to further improve the performance. Rizve et al. [22] explores the complementary strengths of invariant and equivariant representations and self-distillation for better pretraining and performs significantly better than previous methods. Partner-assisted learning [15], which also uses feature level knowledge distillation has proven to be effective.

**Base examples for few-shot learning.** AssoAlign [1] makes use of large base dataset by searching nearest neighbors of the novel examples in the pool of base examples along with adversarial associative alignment. pseudo-labeling is a well-known technique for semi-supervised learning and have been successfully used in several vision tasks, e.g., image and video classification [21]. Jian et al. [11] used pseudo-labeling of the entire base examples and obtained significant performance boost. Afrasiyabi et al. [2] also used mixture-based feature space learning [2] and matching feature sets [3] to further improve the performance.

**Mixup Approaches.** These are simple yet effective regularization techniques for training deep networks. Several variants of mixup have been proposed in the literature, viz., input mixup [39], cutmix [37], manifold-mixup [32], etc. Manifold mixup uses feature level mixup, which provides smoother decision boundary and flattened class representations. Mangla et al. [16] use manifold mixup for few-shot learning from a regularization perspective. In contrast to the above methods, we are using manifold mixup to tackle the data scarcity problem in few-shot learning by generating more samples.

## 3 Problem Statement

We first define the few-shot classification problem addressed in this work. Typically, there exists a large-scale labeled base dataset and novel classes with few samples per class. The task is to learn a feature representation on the base class with abundant samples and then discriminatively learn a representation to recognize subsequent novel classes. Let the base dataset be denoted as $\mathcal{D}^{\text{base}} = \{x_t^{\text{base}}, y_t^{\text{base}}\}_{t=1}^{N^{\text{base}}}$, where $x_t^{\text{base}}$ is the base class sample and the corresponding label is $y_t^{\text{base}} \in C^{\text{base}}$. Similarly, the novel dataset is denoted by $\mathcal{D}^{\text{novel}} = \{x_t^{\text{novel}}, y_t^{\text{novel}}\}_{t=1}^{N^{\text{novel}}}$, where $x_t^{\text{novel}}$ is the novel class sample and the corresponding label is $y_t^{\text{novel}} \in C^{\text{novel}}$. The base and novel classes are disjoint, i.e., $C^{\text{base}} \cap C^{\text{novel}} = \emptyset$, and $|C^{\text{base}}| \geq |C^{\text{novel}}|$.

The training and testing is performed in episodes on the novel class samples. In each episode $i$ of a N-way K-shot problem, the few-shot learner is trained on a support set, i.e., $\mathcal{D}_i^{\text{support}} = \{x_{i,t}^{\text{support}}, y_{i,t}^{\text{support}}\}_{t=1}^{NK}$ for N novel classes containing K samples per class. Then, the few-shot learner is evaluated on the query set $\mathcal{D}_i^{\text{query}} = \{x_{i,t}^{\text{query}}, y_{i,t}^{\text{query}}\}_{t=1}^{NK}$ on the same N classes of $D_i^{\text{support}}$.

## 4 Proposed Methodology

Inspired by the efficacy of pseudo-labeling, we further increase sample size by mixup and choose more informative samples by hard mixup-based sample selection. Our proposed method (FeLMi), outlined in Fig. 1 consists of the following six stages: (i) Learning an embedding on the base dataset ($\mathcal{D}^{\text{base}}$), (ii) Pseudo labeling of the base dataset, (iii) Entropy-based filtering of the pseudo-labeled base dataset, (iv) Mixup sample generation using both base and novel examples, (v) Uncertainty-based hard-mixup sample selection, and (vi) Finetune on the entire dataset using filtered base examples, novel examples and hard-mixed examples. We next describe each of these steps in detail.

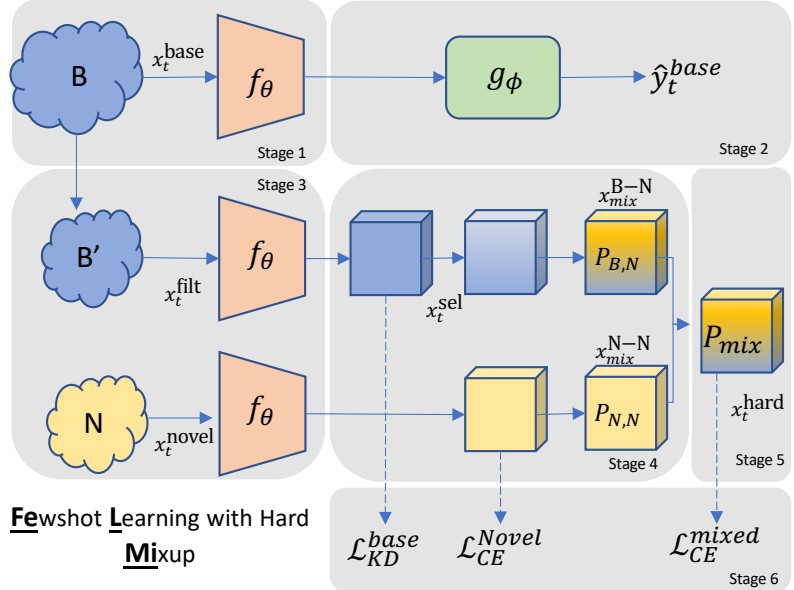

Figure 1: Our proposed method, FeLMi consists of six stages. Stage (1): backbone ($f_\theta$) is pretrained on the base dataset B. Stage (2): pseudo-labels ($\hat{y}_t^{base}$) of the entire base dataset are generated using classifier $g_\phi$. Stage (3): base samples are filtered based on entropy to obtain a reduced dataset B'. Stage (4): novel-novel ($x_{mix}^{N-N}$) and base-novel ($x_{mix}^{B-N}$) mixup samples are generated. Stage (5): hard examples ( $x_t^{hard}$) are selected based on margin. Stage (6): The model is trained on the combination of losses.

### 4.1 Learning embedding on the base dataset

We first learn a discriminative representation from the base dataset $D^{base}$. The convolutional neural network consists of a backbone $f_\theta$ and a final classification layer $h_\phi$. The parameters $(\theta, \phi)$ are jointly learned using the cross-entropy loss on the base dataset $D^{base}$. We use the self-supervised Invariant and Equivariant Represenatation learning (IER) framework by Rizve et al. [22] which makes use of self-supervised losses in addition to standard cross-entropy loss to learn good representations.

$$(\theta^{base}, \phi^{base}) = \text{argmin}_{\theta,\phi} \mathbb{E}_{\{x,y\} \in D^{base}} L_{CE}(h_\phi(f_\theta(x)), y) \tag{1}$$

Large base datasets ensure that we have a good pretrained model to use for meta-testing stage.

### 4.2 pseudo-labeling of base dataset using classifier trained on novel dataset

In this step, we train a logistic regression classifier using the novel examples. Then, using the trained classifier, we generate the pseudo-labels on the entire base dataset following Jian et al. [11]. This pseudo-labeling step partially mitigates the data scarcity problem prevalent in few-shot training. For each episode $i$, we learn a linear classifier $\phi_i$ using the support set of novel examples $D_i^{support}$, such that

$$\phi_i = \text{argmin}_\phi \mathbb{E}_{\{x,y\} \in D_i^{support}} L_{CE}(g_\phi(f_{\theta^{base}}(x)), y). \tag{2}$$

Then, the classifier $g_{\phi_i}$ is used to generate pseudo-labels for the entire base dataset, i.e., $\hat{y}_t^{base} = g_{\phi_i}(f_{\theta^{base}}(x_t))$ for $t = 1, .., N^{base}$.

### 4.3 Entropy based pseudo-label filtering

Given that base classes are disjoint from novel classes, not all classes will provide useful information for the task at hand. Specifically, base samples with high entropy pseudo-labels might lead to confusing samples. Therefore, we perform a simple pseudo-label filtering based on entropy.

$$\mathcal{D}^{base\_filt} = \{\hat{y}_t^{base} | H(\hat{y}_t^{base}) \leq \tau \text{ where } t = 1, \cdots, N^{base}\} \tag{3}$$

We empirically select the entropy threshold $\tau$. After refinement we have $|\mathcal{D}^{\texttt{base\_filt}}|$ base examples. $H(.)$ here denotes the entropy.

## 4.4 Mixup sample generation

Handling the data scarcity problem using pseudo-labels motivates us to generate even more samples using a simple yet an effective mixup strategy. Particularly, we use a feature level manifold mixup step to generate more synthetic samples. Since we have both the base and novel examples, we mix up both base-novel and novel-novel pairs. Note that, since the mixed samples should be close to novel examples, we are not performing base-base mixup. A straightforward way of generating synthetic examples through mixup is to mix two *pure* novel examples. We call this strategy 'Novel-Novel' mixup.

### 4.4.1 Novel-Novel Mixup

We perform manifold mixup on novel-novel samples to create a pool of mixed up samples $P_{N,N}$. We select $\{(x^{\texttt{novel}}, y^{\texttt{novel}}), (\bar{x}^{\texttt{novel}}, \bar{y}^{\texttt{novel}})\} \in D^{\texttt{support}}$, pass them through the feature extractor $f_{\theta^{\texttt{base}}}$ and mixup with $\lambda_n \sim \text{Beta}(\alpha, \alpha)$ to generate $(x_{mix}, y_{mix})$ where:

$$x_{mix}^{N-N} = \lambda_n . f_{\theta^{\texttt{base}}}(x^{\texttt{novel}}) + (1 - \lambda_n) f_{\theta^{\texttt{base}}}(\bar{x}^{\texttt{novel}})$$
$$y_{mix}^{N-N} = \lambda_n . y^{\texttt{novel}} + (1 - \lambda_n) \bar{y}^{\texttt{novel}} \tag{4}$$

The pool of novel-novel mixup samples of size $l$ are generated through randomly sampling $\lambda_n$, i.e., $P_{N,N} = \{(x_{mix}^{N-N}, y_{mix}^{N-N})_i\}_{i=1}^{l}$.

### 4.4.2 Base-Novel Mixup

Access to the base examples during training and their pseudo-labels allows us to make use of these for mixup. The motivation behind using mixup is to generate and augment the small novel set. Since the base classes are disjoint from novel classes and we only have a weak mapping through the pseudo-labels, we employ the following steps while using base examples with mixup. 1) We refrain from mixing two base examples, instead always mixes a base example with a novel example. 2) We only mix with base examples that are close to novel examples. We ensure this by choosing base examples which have a low pseudo-label entropy. 3) We choose a small mixup lambda corresponding to the base examples when mixing them ensuring they remain proximal to the distribution of novel samples. Specifically, we sample $\lambda_b \sim \text{Uniform}(0, 0.2)$. From $(x^{\texttt{base}}, \hat{y}^{\texttt{base}}) \in D^{\texttt{base\_filt}}$, we select k-lowest entropy base examples, i.e., $(x_{sel}^{\texttt{base}}, \hat{y}_{sel}^{\texttt{base}})$, such that,

$$\{(x_{sel}^{\texttt{base}}, \hat{y}_{sel}^{\texttt{base}})\} = \{(x_i, y_i) | i \in \texttt{bottom\_k}(H(\hat{y}))\} \tag{5}$$

Novel examples are $(x^{\texttt{novel}}, y^{\texttt{novel}}) \in D^{\texttt{support}}$ mixed with these selected base examples to generate $(x_{mix}^{B-N}, y_{mix}^{B-N})$ as follows.

$$x_{mix}^{B-N} = \lambda_b . f_{\theta^{\texttt{base}}}(x_{sel}^{\texttt{base}}) + (1 - \lambda_b) f_{\theta^{\texttt{base}}}(x^{\texttt{novel}})$$
$$y_{mix}^{B-N} = \lambda_b . \hat{y}_{sel}^{\texttt{base}} + (1 - \lambda_b) y^{\texttt{novel}} \tag{6}$$

The pool of base-novel mixup samples of size $l$ are generated through randomly sampling $\lambda_b$, i.e., $P_{B,N} = \{(x_{mix}^{B-N}, y_{mix}^{B-N})_i\}_{i=1}^{l}$

## 4.5 Hard mixup sample generation

In the section above, we have generated a pool of base-novel and novel-novel mixed up samples denoted by $P_{B,N}$ and $P_{N,N}$ respectively. Now, we concatenate the generated mixed-up samples, i.e., $P_{mix} = P_{B,N} \cup P_{N,N}$ and then choose hardest $N$ samples based on a uncertainty measure.

More specifically, to get an estimate of hardness of a mixed example chosen from $P_{mix}$, we first obtain a (potentially noisy) estimate of the class probability p $(= g_{\phi_i}(x))$ using the classifier trained in stage 2. We then estimate margin (difference in top-2 probabilities), a measure of uncertainty that is commonly used in active learning [24] and pick mixed samples which have a low margin.

Intuitively, samples with a lower margin lie close to the class boundary thus providing a stronger learning signal.

We choose k-smallest margin values as hard examples (denoted as $P_{hard\_mix}$) chosen from both the base and novel mixed up samples are as follows.

$$\mathcal{P}_{hard\_mix} = \texttt{bottom\_k}\{\texttt{margin}(g_{\phi_i}(f_{\theta^{\texttt{base}}}(x)) \,|\, (x,y) \in P_{mix}\} \tag{7}$$

### 4.6 Finetune on the entire dataset

In the last stage, the model is finetuned on a combined loss computed using the filtered base examples ($\mathcal{D}^{\texttt{base\_filt}}$), the novel examples ($\mathcal{D}^{\texttt{novel}}$), and the hard mixup samples ($\mathcal{P}_{hard\_mix}$). For the base examples, KL divergence loss is used, whereas cross-entropy loss has been used for novel and mixup samples. The final loss is denoted as,

$$\begin{aligned}
\mathcal{L} = \;&\mathbb{E}_{\{x,\hat{y}\}\in\mathcal{D}^{\texttt{base\_filt}}} L_{KD}(g_\phi(f_\theta(x)), \hat{y}) \\
&+ \beta\mathbb{E}_{\{x,y\}\in\mathcal{D}^{\texttt{novel}}} L_{CE}(g_\phi(f_\theta(x)), y) + \gamma\mathbb{E}_{\{x,y\}\in\mathcal{P}_{\texttt{hard\_mix}}} L_{CE}(g_\phi(f_\theta(x)), y)
\end{aligned} \tag{8}$$

$\beta$ and $\gamma$ are scaling parameters for losses corresponding to novel and mixed up samples, respectively.

Finally, the trained model is evaluated on the query set of the episode $i$ ($D_i^{\texttt{query}}$) and averaged accuracy over all the episodes are reported.

## 5 Experimental Results

### 5.1 Experimental Setup

**Datasets:** We experiment on four widely used few-shot benchmarks: FC-100 [17], CIFAR-FS [4], miniImageNet [33] and tieredImageNet [20]. FC-100 is a subset of CIFAR-100, containing 60 classes for meta-training, 20 classes for meta-validation and 20 classes for meta-testing. CIFAR-FS is also obtained from CIFAR-100, containing 64 classes for meta-training, 16 classes for meta-validation and 20 classes for meta-testing. miniImageNet is derived from ImageNet with images downsampled to a resolution of $84\times 84$ pixels. It has 64 classes for meta-training, 16 classes for meta-validation and 20 classes for meta-testing. tieredImageNet [20] is another subset of ImageNet, containing total of 608 classes, from which 351 classes are used for meta-training, 97 classes for meta-validation and 160 classes for meta-testing.

**Implementation details:** For a fair comparison with prior works, we use the ResNet-12 architecture as the backbone network. We attach a two-layer MLP on top of the feature extractor which outputs N-way logits. We use the SGD optimizer with momentum 0.9. Learning rate of backbone and classifier are set to 0.025 and 0.05 respectively with weight decay of 5e-4. Following [11], we use the temperature coefficient of 4.0 for our KD loss. We use a minibatch size of 250 for training. Using standard setting, we also perform data augmentation using color jittering, random crop and horizontal flip. Further details of the hyperparameters and training are provided in the supplementary material.

### 5.2 Results on benchmark datasets

We have compared our proposed method (FeLMi) with state-of-the-art techniques on the benchmark datasets: FC-100, CIFAR-FS, miniImageNet and tieredImageNet. We obtain consistent improvements over state-of-the-art showing the efficacy of our approach on all the aforementioned few-shot benchmarks for both 1-shot and 5-shot settings in Table 1, Table 2 and Table 3.

### 5.3 Results on cross-domain benchmark

We test our approach for more challenging cross-domain settings, where the base-pretraining is done on miniImageNet and the novel classes stem from samples from another domain, e.g., CUB [34]. Our approach achieves significant performance improvements for both the 5-shot and 1-shot setting in this setting as shown in Table 4. Since CUB performs fine-grained classification, we hypothesize that hard mixup samples are even more useful in this case.

Table 1: Comparison of FeLMi (ours) to prior works on CIFAR-FS. Following prior work, we report our results with 95% confidence intervals on meta-testing split of the dataset. § denotes our reproduced numbers using publicly available implementations.

| Model | Backbone | CIFAR-FS 5-way | |
| | | 1-shot | 5-shot |
| --- | --- | --- | --- |
| ProtoNet [28] (NIPS'17) | ResNet-12 | $72.2 \pm 0.7$ | $83.5 \pm 0.5$ |
| MetaOptNet [13] (CVPR'19) | ResNet-12 | $72.6 \pm 0.7$ | $84.3 \pm 0.5$ |
| Shot-Free [19] (ICCV'19) | ResNet-12 | $69.2 \pm$ n/a | $84.7 \pm$ n/a |
| DSN-MR [27] (CVPR'20) | ResNet-12 | $75.6 \pm 0.9$ | $86.2 \pm 0.6$ |
| RFS-simple [31] (ECCV'20) | ResNet-12 | $71.5 \pm 0.8$ | $86.0 \pm 0.5$ |
| RFS-distill [31] (ECCV'20) | ResNet-12 | $73.9 \pm 0.8$ | $86.9 \pm 0.5$ |
| SKD-GEN1 [18] (Arxiv'20) | ResNet-12 | $76.6 \pm 0.9$ | $88.6 \pm 0.5$ |
| IER-distill [22] (CVPR'21) | ResNet-12 | $77.6 \pm 1.0$ | $89.7 \pm 0.6$ |
| PAL [15] (ICCV'21) | ResNet-12 | $77.1 \pm 0.7$ | $88.0 \pm 0.5$ |
| Label-Halluc [11] (AAAI'22) | ResNet-12 | $78.0 \pm 1.0$[§] | $89.37 \pm 0.6$[§] |
| FeLMi | ResNet-12 | $\mathbf{78.22 \pm 0.7}$ | $\mathbf{89.47 \pm 0.5}$ |

Table 2: Comparison of FeLMi (ours) to prior works on FC-100. Following prior work, we report our results with 95% confidence intervals on meta-testing split of the dataset. § denotes our reproduced numbers using publicly available implementations.

| Model | Backbone | FC-100 5-way | |
| | | 1-shot | 5-shot |
| --- | --- | --- | --- |
| ProtoNet [28] (NIPS'17) | ResNet-12 | $37.5 \pm 0.6$ | $52.5 \pm 0.6$ |
| TADAM [17] (NIPS'18) | ResNet-12 | $40.1 \pm 0.4$ | $56.1 \pm 0.4$ |
| MetaOptNet [13] (CVPR'19) | ResNet-12 | $41.1 \pm 0.6$ | $55.5 \pm 0.6$ |
| MTL [29] (CVPR'19) | ResNet-12 | $45.1 \pm 1.8$ | $57.6 \pm 0.9$ |
| DeepEMD [38] (CVPR'20) | ResNet-12 | $46.5 \pm 0.8$ | $63.2 \pm 0.7$ |
| RFS-simple [31] (ECCV'20) | ResNet-12 | $42.6 \pm 0.7$ | $59.1 \pm 0.6$ |
| RFS-distill [31] (ECCV'20) | ResNet-12 | $44.6 \pm 0.7$ | $60.9 \pm 0.6$ |
| AssoAlign [1] (ECCV'20) | ResNet-18 | $45.8 \pm 0.5$ | $59.7 \pm 0.6$ |
| SKD-GEN1 [18] (Arxiv'20) | ResNet-12 | $46.5 \pm 0.8$ | $64.2 \pm 0.8$ |
| InfoPatch [10] (AAAI'21) | ResNet-12 | $43.8 \pm 0.4$ | $58.0 \pm 0.4$ |
| IER-distill [22] (CVPR'21) | ResNet-12 | $48.1 \pm 0.8$ | $65.0 \pm 0.7$ |
| PAL [15] (ICCV'21) | ResNet-12 | $47.2 \pm 0.6$ | $64.0 \pm 0.6$ |
| Label-Halluc [11] (AAAI'22) | ResNet-12 | $47.37 \pm 0.7$[§] | $67.92 \pm 0.7$[§] |
| FeLMi | ResNet-12 | $\mathbf{49.02 \pm 0.7}$ | $\mathbf{68.68 \pm 0.7}$ |

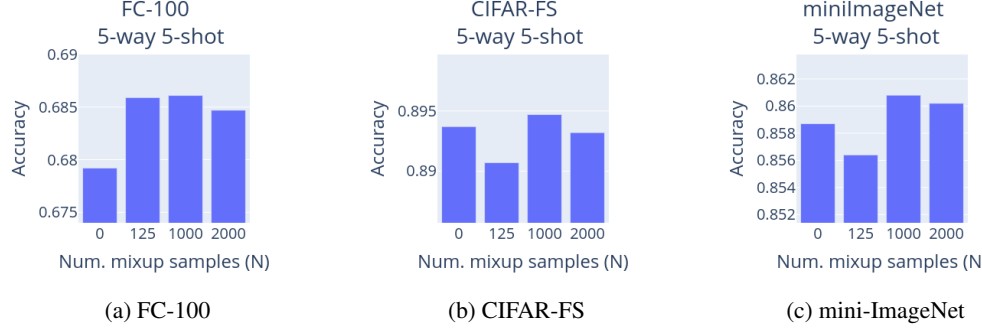

(a) FC-100     (b) CIFAR-FS     (c) mini-ImageNet

Figure 2: Effect of changing number of mixup samples (N). In this experiment we investigate the effect of N on the final cumulative accuracy for the 5-way 1-shot on the three datasets. We notice that N=1000 shows consistent improvements across all datasets.

Table 3: Comparison of our method (FeLMi) against the state-of-the-art on miniImageNet and tiered-ImageNet. Following prior work, we report our results with 95% confidence intervals on meta-testing split of the dataset. § denotes our reproduced numbers using publicly available implementations.

| model | backbone | miniImageNet 5-way | | tieredImageNet 5-way | |
| --- | --- | --- | --- | --- | --- |
| | | 1-shot | 5-shot | 1-shot | 5-shot |
| ProtoNet [28] (NIPS'17) | ResNet-12 | $60.37 \pm 0.83$ | $78.02 \pm 0.57$ | $65.65 \pm 0.92$ | $83.40 \pm 0.65$ |
| TADAM [17] (NIPS'18) | ResNet-12 | $58.50 \pm 0.30$ | $76.70 \pm 0.30$ | - | - |
| TapNet [36] (ICML'19) | ResNet-12 | $61.65 \pm 0.15$ | $76.36 \pm 0.10$ | $63.08 \pm 0.15$ | $80.26 \pm 0.12$ |
| MetaOptNet [13] (CVPR'19) | ResNet-12 | $62.64 \pm 0.61$ | $78.63 \pm 0.46$ | $65.99 \pm 0.72$ | $81.56 \pm 0.53$ |
| MTL [29] (CVPR'19) | ResNet-12 | $61.20 \pm 1.80$ | $75.50 \pm 0.80$ | $65.62 \pm 1.80$ | $80.61 \pm 0.90$ |
| Shot-Free [19] (ICCV'19) | ResNet-12 | $59.04 \pm 0.43$ | $77.64 \pm 0.39$ | $66.87 \pm 0.43$ | $82.64 \pm 0.43$ |
| DSN-MR [27] (CVPR'20) | ResNet-12 | $64.60 \pm 0.72$ | $79.51 \pm 0.50$ | $67.39 \pm 0.83$ | $82.85 \pm 0.56$ |
| DeepEMD [38] (CVPR'20) | ResNet-12 | $65.91 \pm 0.82$ | $82.41 \pm 0.56$ | $71.16 \pm 0.87$ | $86.03 \pm 0.58$ |
| FEAT [35] (CVPR'20) | ResNet-12 | $66.78 \pm 0.20$ | $82.05 \pm 0.14$ | $70.80 \pm 0.23$ | $84.79 \pm 0.16$ |
| Neg-Cosine [14] (ECCV'20) | ResNet-12 | $63.85 \pm 0.81$ | $81.57 \pm 0.56$ | - | - |
| RFS-simple [31] (ECCV'20) | ResNet-12 | $62.02 \pm 0.63$ | $79.64 \pm 0.44$ | $69.74 \pm 0.72$ | $84.41 \pm 0.55$ |
| RFS-distill [31] (ECCV'20) | ResNet-12 | $64.82 \pm 0.82$ | $82.41 \pm 0.43$ | $71.52 \pm 0.69$ | $86.03 \pm 0.49$ |
| AssoAlign [1] (ECCV'20) | ResNet-18 | $59.88 \pm 0.67$ | $80.35 \pm 0.73$ | $69.29 \pm 0.56$ | $85.97 \pm 0.49$ |
| AssoAlign [1] (ECCV'20) | WRN-28-10 | $65.92 \pm 0.60$ | $82.85 \pm 0.55$ | - | - |
| SKD-GEN1 [18] (Arxiv'20) | ResNet-12 | $66.54 \pm 0.97$ | $83.18 \pm 0.54$ | $72.35 \pm 1.23$ | $85.97 \pm 0.63$ |
| P-Transfer [25] (AAAI'21) | ResNet-12 | $64.21 \pm 0.77$ | $80.38 \pm 0.59$ | - | - |
| MELR [8] (ICLR'21) | ResNet-12 | $67.40 \pm 0.43$ | $83.40 \pm 0.28$ | $72.14 \pm 0.51$ | $87.01 \pm 0.35$ |
| IEPT [40] (ICLR'21) | ResNet-12 | $67.05 \pm 0.44$ | $82.90 \pm 0.30$ | $72.24 \pm 0.50$ | $86.73 \pm 0.34$ |
| IER-distill [22] (CVPR'21) | ResNet-12 | $66.85 \pm 0.76$ | $84.50 \pm 0.53$ | $\mathbf{72.71 \pm 0.89}$ | $86.57 \pm 0.81$ |
| Label-Halluc [11](AAAI'22) | ResNet-12 | $67.04 \pm 0.7^{\S}$ | $85.87 \pm 0.48^{\S}$ | $71.97 \pm 0.89$ | $86.80 \pm 0.58$ |
| FeLMi | ResNet-12 | $\mathbf{67.47 \pm 0.78}$ | $\mathbf{86.08 \pm 0.44}$ | $71.63 \pm 0.89$ | $\mathbf{87.07 \pm 0.55}$ |

Table 4: Comparison of FeLMi (ours) in cross-domain setting (miniImageNet $\rightarrow$ CUB). We obtain significant boost compared to prior approaches showing the efficacy of our approach.

| model | backbone | miniImageNet $\rightarrow$ CUB 5-way | |
| --- | --- | --- | --- |
| | | 1-shot | 5-shot |
| Baseline++ [6] (ICLR'19) | ResNet-18 | $40.44 \pm 0.75$ | $56.64 \pm 0.72$ |
| MetaOptNet [13] (CVPR'19) | ResNet-18 | $44.79 \pm 0.75$ | $64.98 \pm 0.68$ |
| S2M2R [16] (WACV'20) | ResNet-18 | $48.24 \pm 0.84$ | $70.44 \pm 0.75$ |
| AssoAlign [1] (ECCV'20) | ResNet-18 | $47.25 \pm 0.76$ | $72.37 \pm 0.89$ |
| MixFSL [2] (ICCV'21) | ResNet-18 | - | $68.77 \pm 0.9$ |
| MT-ConFT [7] (ICCV'21) | ResNet-10 | $49.25 \pm 0.83$ | $74.45 \pm 0.71$ |
| FeLMi | ResNet-12 | $\mathbf{51.66 \pm 0.82}$ | $\mathbf{77.61 \pm 0.69}$ |

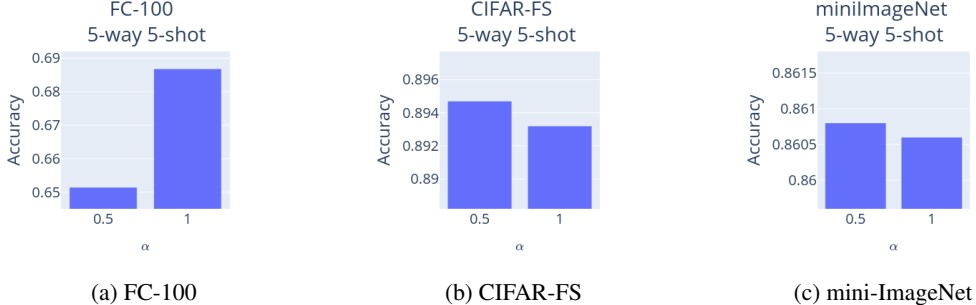

(a) FC-100          (b) CIFAR-FS          (c) mini-ImageNet

Figure 3: Effect of changing number of $\alpha$ parameter of the Beta distribution. In this experiment we investigate the effect of $\alpha$ on the final cumulative accuracy for the 5-way 1-shot on the three datasets. Note that $\alpha$ effectively controls the $\lambda$ values that we sample for N-N mixup. A value of 1 implies sampling from a uniform distribution whereas 0.5 samples $\lambda$ closer to 0 or 1. We notice that $\alpha = 1$ shows consistent improvements across all datasets.

Table 5: Contribution of our proposed techniques to the final performance on FC-100 for 5-way 5-shot classification. We see that each of our proposed techniques leads to an improvement.

| Approach | Accuracy |
|---|---|
| IER [22] | 65.00 |
| + pseudo-label [11] | 67.92 |
| + entropy filtering | 67.96 |
| + Mixup | 68.49 |
| + hard selection | 68.68 |

Table 6: Evaluating different mixup strategies for 5-way 5-shot classification. Our proposed approach of using N-N mixup improves performance. Additionally using B-N mixup leads to further gains.

| Mixup Approach | $\lambda_b$ | $\lambda_n$ | FC-100 | CIFAR-FS |
|---|---|---|---|---|
| B-N + N-N | $U(0, 0.2)$ | $B(1,1)$ | 68.68 | 89.47 |
| B-N + N-N | $B(1,1)$ | $B(1,1)$ | 68.49 | 89.26 |
| N-N | - | $B(1,1)$ | 68.57 | 89.4 |
| None | - | - | 67.92 | 89.37 |

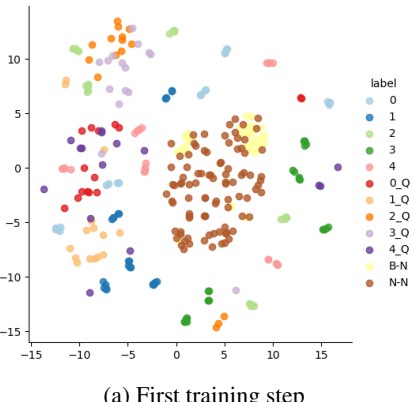

(a) First training step

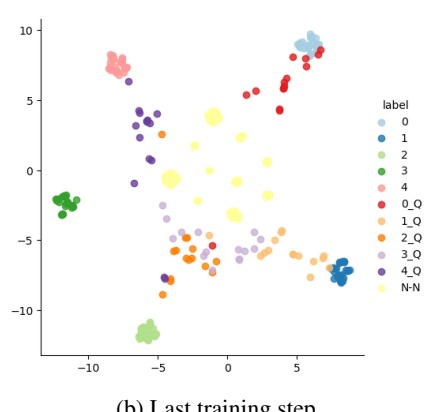

(b) Last training step

Figure 4: tSNE visualizations. We visualize the tSNE plots of the learned representations at the start of training and at the end for one random episode. We see that as training progresses, the data gets more clustered and query labels (denoted by $y\_Q$) get close to the support set clusters. We also overlay the generated mixup samples. These samples offer a good training signal to learn better class boundaries.

## 5.4 Ablation Studies

In this section, we investigate the effect of each stage of our method on the overall performance.

**Ablation of each individual contribution:** We start with the base pretraining similar to IER [22] followed by pseudo-labeling of the entire base dataset akin to Jian et al. [11]. Entropy-based filtering of the pseudo-labels provides an improvement of 0.04%. Using both base and novel sample mixup, we get an improvement of 0.53%. Margin-based hard mixup sample selection additionally provide a performance boost of 0.19% on FC-100 dataset in 5-way 5-shot setting as shown in Table. 5.

**Effect of mixup strategies:** Next, we evaluate different mixup strategies, results are provided in Table. 6 on both FC-100 and CIFAR-FS in 5-way 5-shot settings. In both the cases, we observe that best results are obtained when incorporating base-novel mixup with $\lambda_b \sim$ Uniform(0,0.2) in addition to novel-novel mixup with $\lambda_n \sim$ Beta(1,1). This aforementioned setting ensures the mixed up base-novel samples (denoted by yellow points in Fig. 4 (a)) lie close to the novel examples (labeled 0-4 in Fig. 4 (a)).

**Effect of mixup sample size:** We observe the variation of 5-way 5-shot classification accuracy vs. number of mixup sample size for FC-100, CIFAR-FS and miniImageNet in Fig. 2a, Fig. 2b and Fig. 2c, respectively. We notice that in all the settings, mixup sample size of 1000 performs the best. Increasing the number of mixed up samples seems to lower the performance a bit. We hypothesize this might be due to the increasing imbalance between 'real' and 'generated' points, thus hurting the learning.

**Effect of mixup parameters:** The variation of 5-way 5-shot classification accuracy with respect to mixup parameter $\alpha$ for FC-100, CIFAR-FS and miniImageNet used for performing novel-novel

sample mixup is shown in Fig. 3 . In CIFAR-FS and miniImageNet, $\alpha = 0.5$ seems to perform better. However, in FC-100, we observe the opposite trend.

**tSNE visualizations:** Next, we analyse the tSNE visualizations of the representations learned using our approach. The visualizations (Figure 4) show these for representations at the first step of training (Figure 4(a)) and after the training is completed for one episode (Figure 4 (b)). We observe that the representations for each class get more clustered as the training progresses. Further, representations of the query samples get closer to the corresponding class support samples. Our generated hard mixup samples are close to the query samples as seen in the figure, which provide the necessary training signal to improve the model.

# 6 Limitations

Like most prior approaches, we make use of the base examples when adapting the classifier for novel examples. This requires access to base examples during meta-testing which can be difficult for very large scale datasets. We would like to fix this requirement in future work.

# 7 Conclusions

Few-shot learning is an important computer vision problem. The inherent data scarcity issue makes this problem more challenging. To address this issue, we propose to use hard mixup to synthetically generate mixup samples based on lowest margin based uncertainty measure. To the best of our knowledge, we are the first to propose hard mixup for few-shot learning. Moreover, we also perform hard mixup on the novel-novel and base-novel samples and further improve performance on the benchmark datasets, e.g., FC-100, CIFAR-FS, miniImageNet and tieredImageNet in both the 1-shot and 5-shot settings. Our approach also significantly outperforms the state-of-the-art in cross-domain few-shot (miniImageNet $\rightarrow$ CUB) settings.

# 8 Acknowledgements

The authors AR, AS, and RC were supported by an ONR MURI grant N00014-20-1-2787.

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
