# Supplementary Material for
# FeLMi : Few shot Learning with hard Mixup

**Aniket Roy**
Johns Hopkins University
Baltimore, USA
`aroy28@jhu.edu`

**Anshul Shah**
Johns Hopkins University
Baltimore, USA
`ashah95@jhu.edu`

**Ketul Shah**
Johns Hopkins University
Baltimore, USA
`kshah33@jhu.edu`

**Prithviraj Dhar**
Reality Labs, Meta
Sunnyvale, USA
`prithvirj95@meta.com`

**Anoop Cherian**
MERL
Cambridge, MA
`cherian@merl.com`

**Rama Chellappa**
Johns Hopkins University
Baltimore, USA
`rchella4@jhu.edu`

## 1 Introduction

In this supplementary, we provide additional details, results and visualizations. We list the key sections of the supplementary below.

1. Training details.
2. Pseudocode of the proposed algorithm.
3. Ablation studies supporting our approach.
4. Experiments w.r.t mixup sample number and parameters.
5. Results on variation of Mixup strategies.
6. Visualizations.
7. Societal impact.

## 2 Training details

For the experiments, we used an Nvidia A5000 workstations with 24 GB GPU memory. Details of parameters that yield the best performance for FC100 [3], CIFAR-FS [1], miniImageNet [5] and tieredImageNet [4] are provided in Tab. 1, Tab. 2, Tab. 3 and Tab. 4 respectively.

## 3 Pseudocode of Algorithm

In this section, we provide a pseudocode of our proposed algorithm (mentioned in Sec. 4 in the main paper) in Algorithm. 1.

## 4 Ablation on datasets

In this section, we perform ablation studies for each of the novel component. For 5-way 5-shot setting, results of ablation study for FC100, CIFAR-FS and miniImageNet are provided in Tab. 5. Compared to the baseline [2], each component provides a consistent boost to the performance. Also, as evident

36th Conference on Neural Information Processing Systems (NeurIPS 2022).

Table 1: Hyperparamters for FC-100

| Hyperparameters | 5 way 5 shot | 5 way 1 shot |
|---|---|---|
| Batchsize | 250 | 250 |
| Learning rate of backbone | 0.025 | 0.025 |
| Learning rate of classifier | 0.05 | 0.05 |
| Optimizer | SGD | SGD |
| Momentum | 0.9 | 0.9 |
| Weight decay | 5e-4 | 5e-4 |
| $\beta$ | 1 | 1 |
| $\gamma$ | 1 | 16 |
| Entropy threshold ($\tau$) | 1.55 | 1.55 |
| $\lambda_b$ | $U(0, 0.2)$ | $U(0, 0.2)$ |
| $\lambda_n$ | $Beta(1, 1)$ | $Beta(1, 1)$ |
| Number of hard example ($k$) | 1000 | 125 |

Table 2: Hyperparamters for CIFAR-FS

| Hyperparameters | 5 way 5 shot | 5 way 1 shot |
|---|---|---|
| Batchsize | 250 | 250 |
| Learning rate of backbone | 0.025 | 0.025 |
| Learning rate of classifier | 0.05 | 0.05 |
| Optimizer | SGD | SGD |
| Momentum | 0.9 | 0.9 |
| Weight decay | 5e-4 | 5e-4 |
| $\beta$ | 0.5 | 1 |
| $\gamma$ | 0.1 | 0.1 |
| Entropy threshold ($\tau$) | 1.55 | 1.55 |
| $\lambda_b$ | $U(0, 0.2)$ | $U(0, 0.2)$ |
| $\lambda_n$ | $Beta(0.5, 0.5)$ | $Beta(1, 1)$ |
| Number of hard example ($k$) | 1000 | 125 |

from Tab. 5, our approach of combining all the novel components performs the best. We observe a similar trend for 5-way 1-shot setting as shown in Tab. 6.

## 5 Effect of number of mixup samples and sampling hyperparameters

We analyzed the effect of mixup sample number and $\alpha$ on 5-way 5-shot performance in Fig.2 and Fig.3 of the main paper. Here we supplement that analysis for the 5-way 1-shot case in Fig. 1. We notice that for 5-way 1-shot setting, $N = 125$ performs comparatively better across datasets. Similar analysis w.r.t mixup parameter $\alpha$ has been shown in Fig. 2 and we observe that $\alpha = 1$ performs consistently better across datasets.

## 6 Results on variation of Mixup strategies

In this section we provide detailed analysis of different variants of mixup strategies.

### 6.1 Mixup based on classes

We explore three variants of mixup in our work, i.e., (1) within-class mixup, (2) across-class mixup and (3) random mixup. We analyse each for the case of 5-way 5 shot classification across datasets. Results are provided in Tab. 7. We observe that for CIFAR-FS, all the mixup variants perform quite similarly. But, for FC100 and miniImageNet, *across-class mixup* performs better than all other variants. Across-class mixup helps create more examples near class boundaries thus providing a better training signal.

Table 3: Hyperparamters for miniImageNet

| Hyperparameters | 5 way 5 shot | 5 way 1 shot |
|---|---|---|
| Batchsize | 250 | 250 |
| Learning rate of backbone | 0.025 | 0.025 |
| Learning rate of classifier | 0.05 | 0.05 |
| Optimizer | SGD | SGD |
| Momentum | 0.9 | 0.9 |
| Weight decay | 5e-4 | 5e-4 |
| $\beta$ | 0.5 | 1 |
| $\gamma$ | 0.1 | 1 |
| Entropy threshold ($\tau$) | 1.55 | 1.55 |
| $\lambda_b$ | U$(0, 0.2)$ | U$(0, 0.2)$ |
| $\lambda_n$ | Beta$(0.5, 0.5)$ | Beta$(1, 1)$ |
| Number of hard example ($k$) | 1000 | 125 |

Table 4: Hyperparamters for tieredImageNet

| Hyperparameters | 5 way 5 shot | 5 way 1 shot |
|---|---|---|
| Batchsize | 250 | 250 |
| Learning rate of backbone | 0.025 | 0.025 |
| Learning rate of classifier | 0.05 | 0.05 |
| Optimizer | SGD | SGD |
| Momentum | 0.9 | 0.9 |
| Weight decay | 5e-4 | 5e-4 |
| $\beta$ | 0.5 | 1 |
| $\gamma$ | 0.1 | 1 |
| Entropy threshold ($\tau$) | 1.55 | 1.55 |
| $\lambda_b$ | U$(0, 0.2)$ | U$(0, 0.2)$ |
| $\lambda_n$ | Beta$(0.5, 0.5)$ | Beta$(1, 1)$ |
| Number of hard example ($k$) | 1000 | 125 |

Table 5: Abaltion on proposed components (5way 5 shot). best results shown in **bold**, second best in underline.

| Approach | FC-100 | CIFAR-FS | miniImageNet |
|---|---|---|---|
| LabelHall [2] | 67.92 | 89.37 | 85.87 |
| + entropy filtering | 67.96 | 89.42 | 85.94 |
| + Mixup | 68.49 | 89.45 | 85.95 |
| + hard selection | **68.68** | **89.47** | **86.08** |

Table 6: Abaltion on proposed components (5way 1 shot). best results shown in **bold**, second best in underline.

| Approach | FC-100 | miniImageNet |
|---|---|---|
| LabelHall [2] | 47.37 | 67.04 |
| + entropy filtering | 47.96 | 67.29 |
| + Mixup | 48.52 | 67.41 |
| + hard selection | **49.02** | **67.47** |

**Algorithm 1** FeLMi: Few-shot Learning with hard Mixup

---

**Input:** Base dataset $\mathcal{D}^{\texttt{base}} = \{x_t^{\texttt{base}}, y_t^{\texttt{base}}\}_{t=1}^{N^{\texttt{base}}}$, Novel dataset $\mathcal{D}^{\texttt{novel}} = \{x_t^{\texttt{novel}}, y_t^{\texttt{novel}}\}_{t=1}^{N^{\texttt{novel}}}$, backbone feature extractor $f_\theta$, N-way, K-shot.
**Output:** $\texttt{accuracy}_{\texttt{query}}$
\# Learning embedding on base dataset

$$(\theta^{\texttt{base}}, \phi^{\texttt{base}}) = \text{argmin}_{\theta, \phi} \mathbb{E}_{\{x,y\} \in D^{\texttt{base}}} L_{CE}(h_\phi(f_\theta(x)), y) \tag{1}$$

\# Pseudolabel the base dataset using classifier $\phi_i$

$$\phi_i = \text{argmin}_\phi \mathbb{E}_{\{x,y\} \in D_i^{\texttt{support}}} L_{CE}(g_\phi(f_{\theta^{\texttt{base}}}(x)), y)$$
$$\hat{y}_t^{\texttt{base}} = g_{\phi_i}(f_{\theta^{\texttt{base}}}(x_t)) \tag{2}$$

\# Entropy based pseudolabel filtering

$$\mathcal{Y}^{\texttt{filt}} = \{\hat{y}_t^{\texttt{base}} | H(\hat{y}_t^{\texttt{base}}) \leq \tau \text{ where } t = 1, \cdots, N^{\texttt{base}}\}$$
$$\mathcal{D}^{\texttt{base\_filt}} = \{(x_t^{base}, \hat{y}_t^{base}) | \hat{y}_t^{base} \in \mathcal{Y}^{\texttt{filt}}\} \tag{3}$$

\# Mixup sample generation
\# Novel-Novel mixup generation

$$\{(x^{\texttt{novel}}, y^{\texttt{novel}}), (\bar{x}^{\texttt{novel}}, \bar{y}^{\texttt{novel}})\} \in D^{\texttt{support}}, \lambda_n \sim \text{Beta}(\alpha, \alpha)$$
$$x_{mix}^{N-N} = \lambda_n . f_{\theta^{\texttt{base}}}(x^{\texttt{novel}}) + (1 - \lambda_n) f_{\theta^{\texttt{base}}}(\bar{x}^{\texttt{novel}})$$
$$y_{mix}^{N-N} = \lambda_n . y^{\texttt{novel}} + (1 - \lambda_n) \bar{y}^{\texttt{novel}}$$
$$P_{N,N} = \{(x_{mix}^{N-N}, y_{mix}^{N-N})_i\}_{i=1}^l \tag{4}$$

\# Base-Novel mixup generation

$$(x^{\texttt{base}}, \hat{y}^{\texttt{base}}) \in D^{\texttt{base}}, \lambda_b \sim \text{Uniform}(0, \alpha)$$
$$(x_{sel}^{\texttt{base}}, \hat{y}_{sel}^{\texttt{base}}) = \{(x_i, y_i) | i \in \texttt{bottom\_k}(H(\hat{y}))\}$$
$$x_{mix}^{B-N} = \lambda_b . f_{\theta^{\texttt{base}}}(x_{sel}^{\texttt{base}}) + (1 - \lambda_b) f_{\theta^{\texttt{base}}}(x^{\texttt{novel}})$$
$$y_{mix}^{B-N} = \lambda_b . \hat{y}_{sel}^{\texttt{base}} + (1 - \lambda_b) y^{\texttt{novel}}$$
$$P_{B,N} = \{(x_{mix}^{B-N}, y_{mix}^{B-N})_i\}_{i=1}^l \tag{5}$$

\# Hard sample selection

$$P_{mix} = P_{B,N} \cup P_{N,N}$$
$$\mathcal{P}_{hard\_mix} = \texttt{bottom\_k}\{\text{margin}(g_{\phi_i}(f_{\theta^{\texttt{base}}}(x))) \,|\, (x, y) \in P_{mix}\} \tag{6}$$

\# Finetune the entire model using combined loss

$$\mathcal{L} = \mathbb{E}_{\{x,\hat{y}\} \in \mathcal{D}^{\texttt{base\_filt}}} L_{KD}(g_\phi(f_\theta(x)), \hat{y})$$
$$+ \beta \mathbb{E}_{\{x,y\} \in \mathcal{D}^{\texttt{novel}}} L_{CE}(g_\phi(f_\theta(x)), y) + \gamma \mathbb{E}_{\{x,y\} \in \mathcal{P}_{\texttt{hard\_mix}}} L_{CE}(g_\phi(f_\theta(x)), y) \tag{7}$$

\# Evaluation on the query set

$$\texttt{accuracy}_{\texttt{query}} = \mathbb{E}_{\{x,y\} \in \mathcal{D}^{\texttt{query}}}(f_\theta(x) == y) \tag{8}$$

---

return $\texttt{accuracy}_{\texttt{query}}$

---

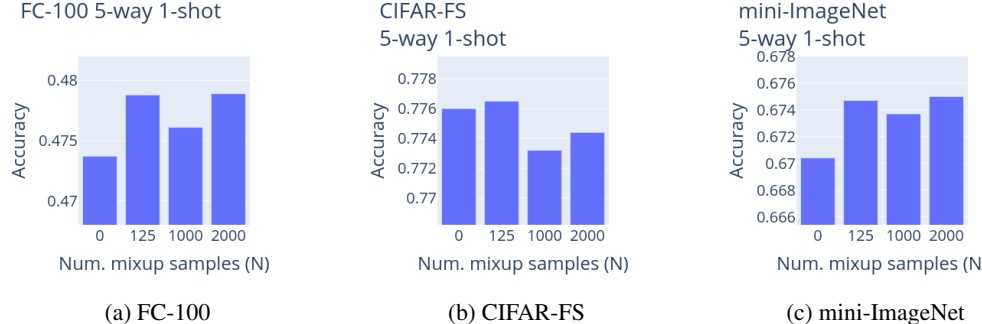

(a) FC-100            (b) CIFAR-FS            (c) mini-ImageNet

Figure 1: Effect of changing number of mixup samples (N). In this experiment, we investigate the effect of N on the final cumulative accuracy for the 5-way 1-shot on the three datasets. We notice that N=125 shows consistent improvements across all datasets.

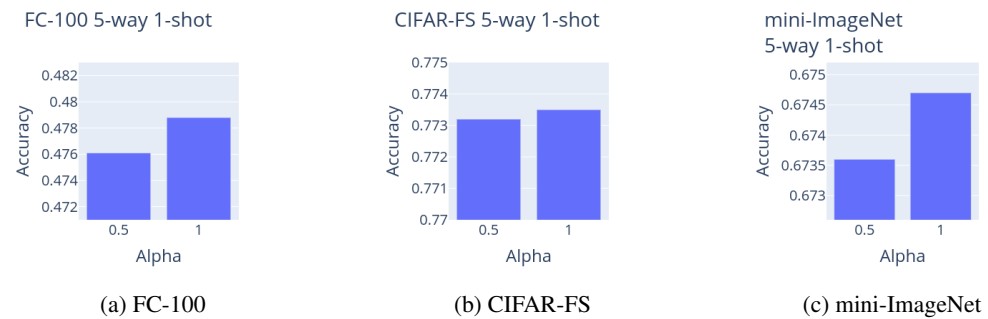

(a) FC-100            (b) CIFAR-FS            (c) mini-ImageNet

Figure 2: Effect of changing number of $\alpha$ parameter of the Beta distribution. In this experiment we investigate the effect of $\alpha$ on the final cumulative accuracy for the 5-way 1-shot on the three datasets. Note that $\alpha$ effectively controls the $\lambda$ values that we sample for N-N mixup. A value of 1 implies sampling from a uniform distribution whereas 0.5 samples $\lambda$ closer to 0 or 1. We notice that $\alpha = 1$ shows consistent improvements across all datasets.

## 6.2   Mixup based on base and novel samples

Another variation of potential mixup would be based on categories of mixup samples, i.e., (1) mixup between only novel-novel examples, and (2) mixup between base-novel and novel-novel examples. The results of our experiments on both of these variants are shown in Tab. 10 for FC100, CIFAR-FS and miniImageNet in 5-way 5-shot settings. As discussed in the main paper in Sec. 4.4, base-novel samples are mixed with parameter $\lambda_b$ and novel-novel samples are mixed with parameter $\lambda_n$. The variants of $\lambda_b$ and $\lambda_n$ and the corresponding 5-way 5-shot performance is also shown in Tab. 10.

To generate samples closer to novel examples during base-novel mixup (Sec. 4.4.2 in the main paper), base examples with bottom-k entropy values are chosen and correspondingly $\lambda_b$ is sampled from uniform distribution Uniform($0, \text{high}_u$). We provide ablation analysis for both $k$ and $high_u$ in Tab. 8 and Tab. 9 respectively in 5-way 5-shot setting.

From Tab. 8, $k = 20$ seems to be a reasonable choice and we perform experiments fixing this value. For miniImageNet, the performance however is quite similar acorss different $k$ values. Ablation on $\text{high}_u$ in Tab. 9 suggests $\text{high}_u = 0.2$ for consistent performance across datasets in 5-way 5-shot setting.

## 7   Visualization

In this section, we provide visualization of our results.

Table 7: Effect of mixup strategies. best results shown in **bold**, second best in underline.

| Approach | FC-100 | CIFAR-FS | miniImageNet |
|---|---|---|---|
| No Mixup | 67.96 | 89.42 | 85.94 |
| Within-class Mixup | 68.18 | **89.48** | 85.74 |
| Random Mixup | 68.62 | 89.47 | 86.07 |
| Across-class Mixup | **68.68** | 89.47 | **86.08** |

Table 8: Ablation on k (bottom-k for base selection during base-novel mixup). best results shown in **bold**, second best in underline.

| k | FC-100 | CIFAR-FS | miniImageNet |
|---|---|---|---|
| 10 | 68.48 | 89.45 | 86.08 |
| 20 | **68.68** | 89.47 | 86.08 |
| 40 | 68.47 | **89.51** | **86.09** |

## 7.1 tSNE visualization

In Fig. 3, we provide the tSNE visualization of the initial and final training and generated mixup samples along with query examples for three random episodes. We notice that the representations are getting more clustered as training progress. Also, the generated hard mixup samples (denoted by yellow points) are close to the query samples, therefore, helps training more generalized model.

## 7.2 Effect of entropy filtering on base samples

[2] used all the base pseudolabeled examples for training. However, pseudolabeling has the inherent problem of generating noisy samples (Fig. 4). In Fig. 4, we visualize the base examples closest (denoted by blue) and farthest (denoted by red) corresponding to the novel examples (denoted by green). We filter out the high entropy pseudolabeled base exmples (noisy samples) by simple entropy thresholding and obtain a small but consistent improvement across shots and datasets as shown in Tab. 5 and Tab. 6. For example, as shown in Fig. 4, novel example class "Malamute" has semantic similarity with closest pseudolabeled base class "Artic Fox", however the farthest base classes, e.g., "solar dish" or "lady bug" do not have any semantic similarity, therefore would be noisy samples. Removing such samples would help the model to learn effectively.

# 8 Societal impact

We do not anticipate any direct negative impact of our work. In fact few-shot learning task is more practical for medical image data, where collecting annotations is difficult. Therefore, learning from small data in the medical domain can have huge positive societal impact.

# References

[1] Luca Bertinetto, Joao F. Henriques, Philip Torr, and Andrea Vedaldi. Meta-learning with differentiable closed-form solvers. In *International Conference on Learning Representations*, 2019.

Table 9: Ablation on $\text{high}_u$ ($\lambda_b$). best results shown in **bold**, second best in underline.

| $\text{high}_u$ | FC-100 | CIFAR-FS | miniImageNet |
|---|---|---|---|
| 0.2 | **68.68** | **89.47** | **86.08** |
| 0.5 | 68.57 | 89.44 | 86.05 |
| 1 | 68.49 | 89.26 | 86.01 |

Table 10: Ablation on Mixup lavels (5-way 5-shot). best results shown in **bold**, second best in underline.

| Mixup Approach | $\lambda_b$ | $\lambda_n$ | FC-100 | CIFAR-FS | miniImageNet |
|---|---|---|---|---|---|
| B-N + N-N | $U(0, 0.2)$ | $B(1, 1)$ | **68.68** | **89.47** | **86.08** |
| B-N + N-N | $B(1, 1)$ | $B(1, 1)$ | 68.49 | 89.26 | 86.01 |
| N-N | - | $B(1, 1)$ | 68.57 | 89.4 | 85.97 |
| None | - | - | 67.92 | 89.37 | 85.87 |

[2] Yiren Jian and Lorenzo Torresani. Label hallucination for few-shot classification. In *Proceedings of the AAAI Conference on Artificial Intelligence*, 2022.

[3] Boris N. Oreshkin, Pau Rodríguez López, and Alexandre Lacoste. Tadam: Task dependent adaptive metric for improved few-shot learning. In *NeurIPS*, 2018.

[4] Mengye Ren, Sachin Ravi, Eleni Triantafillou, Jake Snell, Kevin Swersky, Josh B. Tenenbaum, Hugo Larochelle, and Richard S. Zemel. Meta-learning for semi-supervised few-shot classification. In *International Conference on Learning Representations*, 2018.

[5] Oriol Vinyals, Charles Blundell, Timothy Lillicrap, koray kavukcuoglu, and Daan Wierstra. Matching networks for one shot learning. In *Advances in Neural Information Processing Systems*, 2016.

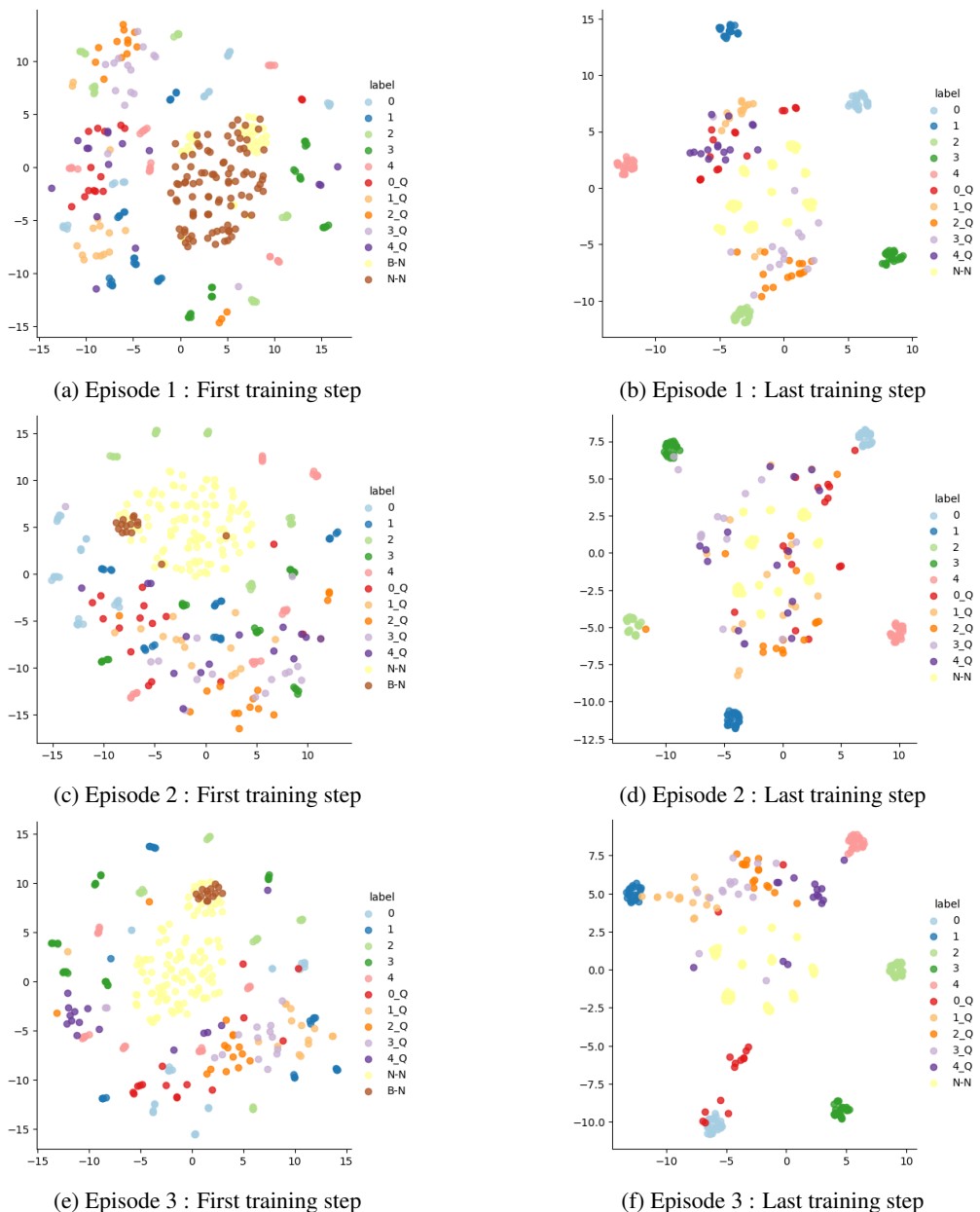

Figure 3: tSNE visualizations. We visualize the tSNE plots of the learned representations at the start of training and at the end for three random episode. We see that as training progresses, the data gets more clustered and query labels (denoted by $y\_Q$) get close to the support set clusters. We also overlay the generated mixup samples. These samples offer a good training signal to learn better class boundaries.

Figure 4: Effect of entropy filtering. We visualize base examples closest (denoted by blue) and farthest (denoted by red) corresponding to the novel examples (denoted by green) based on entropy of the pseudolabels. We discard the farthest (red) base samples during entropy filtering which donot have any semantic similarity w.r.t the novel samples (step 2 in our approach) to train the model effectively. For examples, novel example class "Malamute" has semantic similarity with the closest pseudolabeled base class "Arctic Fox", but do not have any semantic similarity with the farthest base classes like "solar dish" or "lady bug".