# OpenReview forum: "FeLMi : Few shot Learning with hard Mixup"
_NeurIPS.cc/2022/Conference — NeurIPS 2022 Accept_

### Official Review · Reviewer_WHpk · 2022-06-25

**Rating:** 5
**Confidence:** 4
**Soundness:** 3 good
**Presentation:** 3 good
**Contribution:** 2 fair

**Summary:**

In this paper, Few shot Learning with hard Mixup (FeLMi) is developed to mitigate the issue of data scarcity. The proposed method is composed of 6 steps. (1) Pretrain the model on the base dataset using the cross-entropy loss with auxiliary self-supervised loss. (2) Train a linear logistic regression model on the top of the learned feature extractor using the novel data to generate pseudolabels for the base dataset. (3) Filter the pseudolabels based on entropy. (4) Do novel-novel and base-novel mixup to generate more data. (5) Choose hard mixup samples based on the margins in classification probability. (6) Finetune the model using the filtered base data, novel data, and hard mixup data. Experiments indicate that the proposed method leads to improved performance in few-shot learning.

**Questions:**

According to Table 5, base-novel mixup leads to negligible performance improvement. I wonder if this step is necessary for your proposed method.

**Limitations:**

Compared with the simplest fine-tuning methods in few-shot learning (e.g. RFS-simple, ECCV 2020), the proposed method is more computationally expensive. Specifically, the proposed method must assign pseudo labels for the base data. It becomes an issue when the base dataset is huge. It’s better to provide an empirical or theoretical analysis of the time and space complexity.

**Strengths And Weaknesses:**

Strengths:

The paper is well organized and easy to follow.

The proposed method is straightforward and easy to understand.

Sufficient ablation studies in the experiments.


Weaknesses:

Among the 6 steps in FeLMi, only Step 5 is original (choosing hard mixup samples). However, Table 4 indicates that Step 5 only leads to marginal improvement in few-shot classification accuracy.

The model is pretrained by cross-entropy loss and self-supervised loss. It is known that pretraining with auxiliary self-supervised loss leads to improved performance in transfer learning for downstream tasks. Although FeLMi achieves the best performance in Tables 1, 2, and 3, it relies on a stronger pretraining method. It does not demonstrate the effectiveness of hard mixup.

---

> ### Author Response · Authors · 2022-08-02
> **Thanks for the feedback. Comments are below.**
>
> **Q1. [Among the 6 steps in FeLMi, only Step 5 is original (choosing hard mixup samples). However, Table 4 indicates that Step 5 only leads to marginal improvement in few-shot classification accuracy. ]**
>
> Ans: We would like to point out that actually steps 3,4 and 5 have our novel contributions. Also, combining these simple methods would lead to state-of-the-art performance in FSL for various datasets both in in-domain and cross-domain settings. The performance improvements in cross-domain settings are quite significant.
>
> **Q2. [Although FeLMi achieves the best performance in Tables 1, 2, and 3, it relies on a stronger pretraining method. It does not demonstrate the effectiveness of hard mixup.]**
>
> Ans: We would like to point out that we are using the same model as of LabelHaluc[9], therefore the performance improvement compared to LabelHal[9] is solely the effect of hard mixup as shown in Table 4 in the main paper.
>
>
> **Q3. [According to Table 5, base-novel mixup leads to negligible performance improvement. I wonder if this step is necessary for your proposed method.]**
>
> Ans: The improvement of using base-novel mixup might be marginal but consistent across all the settings (Table 9 in supplementary material). Further, since we have access to base examples during training, B-N mixup comes with minimal overheads.
>
> **Q4. [It’s better to provide an empirical or theoretical analysis of the time and space complexity.]**
>
> Ans:
>
> a.**Time complexity**: Since we are doing the pseudolabeling on base examples using a linear classifier, which is of linear time complexity (O(B), B = number of base class) and other methods like entropy filtering and margin based hard mixup sample selection is also of linear time complexity. Therefore the overall time complexity is linear in time.
>
> b.	**Space complexity:** Given the number of base samples (B) and novel samples (N), the total space required would be just O(B+N), since entropy filtering, hard-mixup does not use any additional memory.

---

> ### Comment · Reviewer_WHpk · 2022-08-03
> **Comments to the rebuttal**
>
> Thank you for the reply from the authors.
>
> I do not agree that Steps 3 and 4 are novel. Step 3 is just a self-labeling process, which assigns the label based on high-confidence (or low entropy) prediction [1, 2, 3, 4]. This technique has been well studied in different applications before the era of deep learning.  Those two well-known papers from 2020 should have been discussed in Related Work.
>
> [1] Scudder, Henry. "Probability of error of some adaptive pattern-recognition machines." IEEE Transactions on Information Theory 11.3 (1965): 363-371.
>
> [2] McLachlan, Geoffrey J. "Iterative reclassification procedure for constructing an asymptotically optimal rule of allocation in discriminant analysis." Journal of the American Statistical Association 70.350 (1975): 365-369.
>
> [3] Van Gansbeke, Wouter, et al. "Scan: Learning to classify images without labels." European conference on computer vision. 2020.
>
> [4] Sohn, Kihyuk, et al. "Fixmatch: Simplifying semi-supervised learning with consistency and confidence." Advances in neural information processing systems 33 (2020): 596-608.
>
>
> Step 4 is just mix-up step. It has been used in [5]. You may want to argue that [5] uses mix-up as regularization, while your method uses mix-up to address the data scarcity problem. However, [6] already showed that mix-up is a form of regularization via theoretical analysis. Interpreting mix-up as generating more data is just an intuitive explanation of this type of regularization.
>
> [5] Mangla, Puneet, et al. "Charting the right manifold: Manifold mixup for few-shot learning." Proceedings of the IEEE/CVF winter conference on applications of computer vision. 2020.
>
> [6] Verma, Vikas, et al. "Manifold mixup: Better representations by interpolating hidden states." International Conference on Machine Learning. PMLR, 2019.
>
> In the few-shot learning literature, the standard deviation is reported along with the accuracy. When the accuracy gap between two methods is obviously larger than the standard deviation, one method is claimed to be better than the other method without using rigorous statistical tests. In your case, the gap between the accuracy is small (especially in Table 5). It is necessary to do statistical tests to show that the higher accuracy of one method (or one variant of the method) is not due to the randomness in data.

---

> > ### Author Response · Authors · 2022-08-08
> > **Response to Reviewer WHpk**
> >
> > We thank the esteemed reviewer for the useful comments and suggestions.
> >
> >
> > **Q1. I do not agree that Steps 3 and 4 are novel. Step 3 is just a self-labeling process, which assigns the label based on high-confidence (or low entropy) prediction [1, 2, 3, 4]. This technique has been well studied in different applications before the era of deep learning. Those two well-known papers from 2020 should have been discussed in Related Work.**
> >
> >
> > Ans: We thank the reviewer for the insightful references. We agree that entropy based self-labeling has been used in several computer vision tasks, e.g., Fix-match [4] used entropy based filtering in semi-supervised learning and obtained significant performance boost. Scan[3] on the other hand use self-labeling using pseudolabels in unsupervised image classification. Earlier also entropy based filtering has been found useful in iterative reclassification [1,2].
> > We will definitely cite these and other mentioned papers in the final version.
> >
> > We also want to highlight that we have used simple entropy based filtering to refine the pseudolabels, which has never been explored for few-shot learning to the best of our knowledge. Therefore, the novelty is to apply combination of these useful techniques in the context of few-shot learning. In step 4, we have used mixup between base-novel and novel-novel samples and further improve the performance using hard mixup, which we are the first to propose to the best of our knowledge in few-shot learning. Each of these components might be simple, but we show that combining those could be effective in the context of few-shot learning is the novel contribution of the paper.
> >
> >
> > **Q2. Step 4 is just mix-up step. It has been used in [5]. You may want to argue that [5] uses mix-up as regularization, while your method uses mix-up to address the data scarcity problem. However, [6] already showed that mix-up is a form of regularization via theoretical analysis. Interpreting mix-up as generating more data is just an intuitive explanation of this type of regularization.**
> >
> > Ans: We would like to point out that [5] uses mix-up as a regularization with respect to base classes only. During the pretraining step, they are using manifold mixup using the base class samples only, and also not generating new samples. Therefore, they are not including the novel examples for mixup in their pipeline.
> >
> > On the other hand, we have the freedom to mix-up both novel-novel or base-novel samples to generate more informative samples to combat the data-scarcity problem. Specifically, we have noticed that mixing novel-novel examples provide better results. While, we accept this could be some form of regularization, we claim that using mixup this way provides better regularization as evident from the comparison w.r.t [5] in Table 3 below.
> >
> > **Table 3. Comparison between S2M2R and our method.**
> >
> > | Method     | miniImageNet | miniImageNet | CIFAR-FS   | CIFAR-FS   |
> > |------------|--------------|--------------|------------|------------|
> > |            | 1-shot       | 5-shot       | 1-shot     | 5-shot     |
> > | S2M2R [5] | 64.93 $\pm$ 0.18   | 83.18 $\pm$ 0.11   | 74.81 $\pm$ 0.19 | 87.47 $\pm$ 0.13 |
> > | FeLMi      | **67.47 $\pm$ 0.78**   | **86.08 $\pm$ 0.44**   | **78.22 $\pm$ 0.7**  | **89.47 $\pm$ 0.5**  |
> >
> >
> >
> > **Q3. In the few-shot learning literature, the standard deviation is reported along with the accuracy. When the accuracy gap between two methods is obviously larger than the standard deviation, one method is claimed to be better than the other method without using rigorous statistical tests. In your case, the gap between the accuracy is small (especially in Table 5). It is necessary to do statistical tests to show that the higher accuracy of one method (or one variant of the method) is not due to the randomness in data.**
> >
> > Ans: We thank the reviewer for this question. We have followed the general pratice of reporting the mean and standard deviation of the results in the few-shot learning literature. However, we agree that the statistical significance tests will provide more reliable results. To this end, we have performed Welch's t-test [6] on the experiments and observed that compared to basline LabelHal[9], our results are statistically significant for a p-value of 10% and holds for Table 1, Table 2, Table 3, Table 4 and Table 5. These statistical tests confirm the superiority of our method compared to the baselines. We will include the results in the final version of the paper.
> >
> > [6] Welch, B. L. (1947). "The generalization of "Student's" problem when several different population variances are involved". Biometrika. 34 (1–2): 28–35. doi:10.1093/biomet/34.1-2.28. MR 0019277. PMID 20287819

---

> > > ### Author Response · Authors · 2022-08-09
> > > **Response to Reviewer WHpk**
> > >
> > > Additionally, we would like to draw the attention of the reviewer to the fact that we have performed cross-domain few-shot learning experiments and demonstrate state-of-the-art results on miniImageNet -> CUB dataset (which is considered to be more challenging) and the results are also statistically significant by a large margin. These experiments validate the efficacy of the proposed approach compared to state-of-the-art methods. We envision that using pseudolabel filtering and hard-mixup we can get better domain generatlization of the few-shot learner, which helps for cross-domain few-shot learning as shown below in Table 2.
> > >
> > >
> > > **Table 2: Results for miniImageNet -> CUB**
> > >
> > > | Method     | Backbone  | 1-shot     | 5-shot     |
> > > |------------|-----------|------------|------------|
> > > | Baseline++ | ResNet-18 | 40.44 $\pm$ 0.75 | 56.64 $\pm$ 0.72 |
> > > | MetaOPt [11]  | ResNet-18 | 44.79 $\pm$ 0.75 | 64.98 $\pm$ 0.68 |
> > > | S2M2R [14]     | ResNet-18 | 48.24 $\pm$ 0.84 | 70.44 $\pm$ 0.75 |
> > > | AssoAlign [1] | ResNet-18 | 47.25 $\pm$ 0.76 | 72.37 $\pm$ 0.89 |
> > > | MixFSL     | ResNet-18 | -          | 68.77 $\pm$ 0.9  |
> > > | MT-ConFT   | ResNet-10 | 49.25 $\pm$ 0.83 | 74.45 $\pm$ 0.71 |
> > > | FelMi (Ours)    | ResNet-12 | **51.66 $\pm$ 0.82** | **77.61 $\pm$ 0.69** |
> > >
> > >
> > > **References:**
> > >
> > > [Baseline++] Chen et al. " A closer look at few-shot classification". International Conference on Learning Representations, 2019
> > >
> > > [MixFSL] Afrasiyabi et al. "Mixture-based feature space learning for few-shot Â image classification." International Conference on Computer Vision, 2021
> > >
> > > [MT-ConFT] Das et al. "On the Importance of Distractors for Few-Shot Classification ", International Conference on Computer Vision, 2021

---

### Official Review · Reviewer_2Qz6 · 2022-07-11

**Rating:** 5
**Confidence:** 4
**Soundness:** 3 good
**Presentation:** 2 fair
**Contribution:** 2 fair

**Summary:**

This paper presents FeLMi, a type of mixup method for few-shot learning, with margin-based uncertainty criteria. The work aims to augment new data in a mixup form to tackle the overfitting problem in a few-shot classification setting. With this, the presented method consists of six phases. Overall, after pretraining on the base classes and obtaining the pseudo labels of the base examples, both novel-novel and base-novel mixup samples are generated for data augmentation. The pretraining method employs self-supervised-based Invariant and Equivariant Representation learning (IER) [19]. Finally, the evaluation is presented using three few-shot learning benchmarks.

**Questions:**

- While understanding the use of standard transfer learning instead of an episodic training approach, I would like to see more discussion on why fine-tuning and why not episodic pre-training?
- What about standard pretraining without using a self-supervised [19] training method?
- What about the time complexity of the method?
- Could you please elaborate more on “k-smallest margin values as hard examples” (line 194)?
- What about the fine-tuning implementation details? Also, do the literature uses MLP on top of ResNet12? (“we train a two layer MLP” ll. 214). If so, please cite.
- Why “temperature coefficient” is set to 4.0 (line 217)? Is this obtained through cross-validation?

**Limitations:**

Though the presented method is proposed for few-shot learning, it is only evaluated with image classification. Additionally, evaluation with large datasets (such as Tieredimagenet) is missing. While the presentation of the paper is good, I think the author(s) can improve the related work by clearly discussing the difference between the approaches (such as Base examples for few-shot learning) with the proposed method. I think the conclusion can include some limitations and future works too.

**Strengths And Weaknesses:**

In general, the idea of the paper is interesting. Specifically, I like how the presented method is inspired by several recent works of base examples and mixtup method for few-shot image classification to present novel data-augmentation methods. Additionally, the paper is well-written and easy to follow. However, I think the method contains the following weaknesses:
- The overall model is complex (having several stages with different criteria).
- The method uses many tricks, such as self-supervised learning for pretraining and active learning for the hard mixtup. Therefore, the method's evaluation and justification became more complicated compared to standard methods such as ProtoNet.
- The proposed method is evaluated on three small datasets, but I think some large datasets are required.
- Having a 5-way classification problem, the FeLMi can not gain significant accuracy in CIFAR-FS dataset. I think this is ok, but some extra evaluation might help us understand the proposed method's classification gain/loss.

---

> ### Author Response · Authors · 2022-08-02
> **Thanks for the valuable feedback. Comments are below.**
>
> **Q1. [model is complex]**
>
> Ans:  While our approach consists of multiple steps, each of those is simple and effective (also mentioned by reviewer WHpk, 3CtS). All our steps can be implemented with a few lines of code.
>
> **Q2. [The method uses many tricks, such as self-supervised learning for pretraining and active learning for the hard mixtup. Therefore, the method's evaluation and justification became more complicated compared to standard methods such as ProtoNet.]**
>
> Ans:  Self-supervised learning has shown to be very effective in a lot of different tasks. Few-shot learning is no exception and SSL has shown consistent improvements on various FSL benchmarks [9, 19]. Our approach falls under the subset of works which approach FSL as a transfer learning problem [9, 19]. While the approach is different from ProtoNet, it uses the exact same dataset and protocols. Further, we would like to clarify that our approach does not use active learning. Use of margin (which is the difference between top-1 and top-2 probabilities) is merely inspired from the active learning literature as a measure of uncertainty. Our experiments clearly demonstrate that incorporating even such simple techniques lead to state-of-the-art performances.
>
> **Q3. [I think some large datasets are required.]**
>
> Ans: Thanks for the suggestion. We have included results for two other tasks. (1) Results on TieredImageNet, (2) Cross-domain FSL from miniImageNet to CUB dataset. We have noticed that our methods provide better performance in TieredImageNet dataset and significant performance boost in cross-domain settings (miniImageNet  -> CUB).
>
> **Table 1: Results on tieredImageNet (5-way 5-shot)**
>
> | Method      | Backbone  | 5-way 5shot |
> |-------------|-----------|-----------------|
> | ProtoNet [25]   | ResNet-12 | 83.40 $\pm$ 0.65    |
> | TapNet  [32]    | ResNet-12 | 80.26 $\pm$ 0.12    |
> | MetaOpt  [11]   | ResNet-12 | 81.56 $\pm$ 0.53    |
> | MTL     [26]    | ResNet-12 | 80.61 $\pm$ 0.90    |
> | Shotfree [17]   | ResNet-12 | 82.64 $\pm$ 0.43    |
> | DSN-MR   [24]   | ResNet-12 | 82.85 $\pm$ 0.56    |
> | DeepEMD   [34]  | ResNet-12 | 86.03 $\pm$ 0.58    |
> | FEAT    [31]    | ResNet-12 | 84.79 $\pm$ 0.16    |
> | RFS-Simple [28]  | ResNet-12 | 84.41 $\pm$ 0.55    |
> | RFS-distill [28]| ResNet-12 | 86.03 $\pm$ 0.49     |
> | AssoAlign [1]  | ResNet-18 | 85.97 $\pm$ 0.49    |
> | SKD-Gen1 [16]  | ResNet-12 | 86.61 $\pm$ 0.59    |
> | InfoPatch [8]  | ResNet-12 | 85.44 $\pm$ 0.35    |
> | IEPT     [36]   | ResNet-12 | 86.73 $\pm$ 0.34    |
> | IER-distill [19] | ResNet-12 | 86.57 $\pm$ 0.81    |
> | LabelHalluc [9] | ResNet-12 | 86.80 $\pm$ 0.58    |
> | FeLMi (Ours)      | ResNet-12 | **87.07 $\pm$ 0.55**    |
>
>
> **Table 2: Results for miniImageNet -> CUB**
>
> | Method     | Backbone  | 1-shot     | 5-shot     |
> |------------|-----------|------------|------------|
> | Baseline++ | ResNet-18 | 40.44 $\pm$ 0.75 | 56.64 $\pm$ 0.72 |
> | MetaOPt [11]  | ResNet-18 | 44.79 $\pm$ 0.75 | 64.98 $\pm$ 0.68 |
> | S2M2R [14]     | ResNet-18 | 48.24 $\pm$ 0.84 | 70.44 $\pm$ 0.75 |
> | AssoAlign [1] | ResNet-18 | 47.25 $\pm$ 0.76 | 72.37 $\pm$ 0.89 |
> | MixFSL     | ResNet-18 | -          | 68.77 $\pm$ 0.9  |
> | MT-ConFT   | ResNet-10 | 49.25 $\pm$ 0.83 | 74.45 $\pm$ 0.71 |
> | FelMi (Ours)    | ResNet-12 | **51.66 $\pm$ 0.82** | **77.61 $\pm$ 0.69** |
>
> **References:**
>
> [Baseline++] Chen et al. " A closer look at few-shot classification". International Conference on Learning Representations, 2019
>
> [MixFSL] Afrasiyabi et al. "Mixture-based feature space learning for few-shot Â image classification." International Conference on Computer Vision, 2021
>
> [MT-ConFT] Das et al. "On the Importance of Distractors for Few-Shot Classification ", International Conference on Computer Vision, 2021
>
>
> We observe a significant gain in performance for the cross-domain settings.
>
>
>
> **Q4. [  Having a 5-way classification problem, the FeLMi can not gain significant accuracy in CIFAR-FS dataset. I think this is ok, but some extra evaluation might help us understand the proposed method’s classification gain/loss]**
>
> Ans: We have added more analysis to show the effectiveness of FeLMi. Additionally, we have added results on TieredImageNet and cross-domain few-shot setting in Table 1 and Table 2 respectively.
>
> **Q5. [While understanding the use of standard transfer learning instead of an episodic training approach, I would like to see more discussion on why fine-tuning and why not episodic pre-training?]**
>
> Ans: RFS [28] has shown the finetuning performs better than episodic training and recent other approaches [19, 9] also have verified this. Therefore, we also use transfer learning approach in our case.
>
> More comments are below.

---

> > ### Author Response · Authors · 2022-08-02
> > **More comments**
> >
> > **Q6. [What about standard pretraining without using a self-supervised [19] training method?]**
> >
> > Ans: Our method is generic and can be applied to any pretraining method with/without self-supervised training. But recent literature has shown that self-supervised pretraining is better than standard pretraining. Therefore, we have chosen self-supervised training for pre-training.
> >
> >
> > **Q7. [What about the time complexity of the method?]**
> >
> > Ans: Since we do the pseudolabeling on base examples using a linear classifier, which is of linear time complexity (O(B), B = number of base class) and other methods like entropy filtering and margin based hard mixup sample selection is also of linear time complexity, therefore, the overall time complexity is linear in time.
> >
> > **Q8. [Could you please elaborate more on “k-smallest margin values as hard examples” (line 194)?]**
> >
> > Ans: We use margin as an uncertainty measure. Margin is a uncertainty measure, which is expressed as the difference between top-2 probabilities. Small margin implies less uncertainty, and we choose k-smallest margin values examples, (those are less uncertain) and act as hard example.
> >
> > **Q9. [What about the fine-tuning implementation details? Also, do the literature uses MLP on top of ResNet12? (“we train a two layer MLP” ll. 214). If so, please cite.]**
> >
> > Ans:  Please refer to the supplementary for fine-tuning hyperparameters (Supplementary: Section 2). Yes, the literature does use a 2 layer MLP for the final layer [28, 9].
> >
> > **Q10. [ Why “temperature coefficient” is set to 4.0 (line 217)? Is this obtained through cross-validation? ]**
> >
> > Ans: We did not tune this hyperparameter and it was set as 4.0 following prior work [9].
> >
> > **Q11. [it is only evaluated with image classification].**
> >
> > Ans: In this work we mainly focus on image classification task. However, we believe that this approach is quite generic and can also be applied to other vision (e.g., detection ) and NLP tasks.
> >
> > **Q12. [author(s) can improve the related work..I think the conclusion can include some limitations and future works too]**
> >
> > Ans: Thanks for the suggestion. We will improve the related works section and conclusion based on your feedback in the final version.

---

> > > ### Comment · Reviewer_2Qz6 · 2022-08-10
> > > **Thank you for the rebuttal**
> > >
> > > I read the rebuttal and appreciate the author's (s) effort on addressing the raised concerns. Though I still have some concerns about the complexity of the model and emplying self-supervised learning, my concerns related to the evaluation of the method are addressed. Therefore, I change my score from borderline reject to borderline accept.

---

> > > > ### Author Response · Authors · 2022-08-10
> > > > **Response to reviewer 2Qz6**
> > > >
> > > > We thank the reviewer for appreciating our effort. We would like to clarify some more concerns that the reviewer might have.
> > > >
> > > > **Q1. [Complexity of the model]**
> > > >
> > > > Ans: We agree that our proposed technique has several components. However, we want to point out that the components e.g., entropy filtering, hard mixup are quite simple and easy to implement, yet we observe state-of-the-art performance. Additionally, we have performed rigorous time and space complexity analysis below.
> > > >
> > > > **a. Time complexity**: Since we are doing the pseudolabeling on base examples using a linear classifier, which is of linear time complexity (O(B), B = number of base class) and other methods like entropy filtering and margin based hard mixup sample selection is also of linear time complexity. Therefore the overall time complexity is linear in time.
> > > >
> > > > **b. Space complexity:** Given the number of base samples (B) and novel samples (N), the total space required would be just O(B+N), since entropy filtering, hard-mixup does not use any additional memory.
> > > >
> > > > This analysis shows our proposed technique is simple, yet effective.
> > > >
> > > > **Q2. [emplying self-supervised learning]**
> > > >
> > > > Ans: We thank the reviewer for the question. Traditionally, few-shot learning problem was tackled with meta-learning framework [1]. However, recently, Isola et. al. [2] has shown that incorporating self-supervised learning and transfer learning provides better results on few-shot learning tasks. Recent techniques [3, 4] has also observed this trend. We also have verified this claim and therefore built on transfer learning based method using self-supervised learning and demonstrate state-of-the-art results.
> > > >
> > > >
> > > > We would like to clarify if the reviewer have any more concerns. Thank you!
> > > >
> > > > **References:**
> > > >
> > > > [1] Kwonjoon Lee, Subhransu Maji, Avinash Ravichandran, and Stefano Soatto. ''Meta-learning with differentiable convex optimization.'' In Proceedings of the IEEE/CVF Conference on Computer Vision and Pattern Recognition (CVPR), 2019.
> > > >
> > > >
> > > > [2] Yonglong Tian, Yue Wang, Dilip Krishnan, Joshua B. Tenenbaum, and Phillip Isola. ''Rethinking few-shot image classification: A good embedding is all you need?'' In Andrea Vedaldi, Horst Bischof, Thomas Brox, and Jan-Michael Frahm, editors, Computer Vision – ECCV 2020, 2020
> > > >
> > > > [3] Yiren Jian and Lorenzo Torresani. ''Label hallucination for few-shot classification.'' In Proceedings of the AAAI Conference on Artificial Intelligence, 2022.
> > > >
> > > > [4] Mamshad Nayeem Rizve, Salman Khan, Fahad Shahbaz Khan, and Mubarak Shah. ''Exploring complementary strengths of invariant and equivariant representations for few-shot learning.'' ICCV 2021

---

### Official Review · Reviewer_3CtS · 2022-07-11

**Rating:** 3
**Confidence:** 4
**Soundness:** 1 poor
**Presentation:** 2 fair
**Contribution:** 2 fair

**Summary:**

The paper presents a hard manifold mix up process to augment the few-shot samples during fine-tuning for improving accuracy performance. The mix-up is carried out in different settings: novel-novel, novel-base and the hard samples from the mix up based on margin are utilized during fine-tuning.

**Questions:**

1.  The paper says that the pretraining is done with  self supervised loss in addition to cross entropy, but the eq. 1 just shows cross entropy where is the full loss according to the description?
2. How exactly the entropy threshold for filtering pseudolabels for base classes is selected?
3. How this method is compared with the "Charting the Right Manifold: Manifold Mixup for Few-shot Learning" paper by Mangla et al. in WACV 2020?
4. Why the other standard benchmarks such as tiered-Imagenet or CUB is missing also different architechtures such as WRN? Also, are the methods compared are evaluated based on the same pretrained model? Over, how many episodes the average accuracies are reported?
5. What about domain-shift results for example mini to CUB setting as in previous papers such as "A closer look at few-shot learning" by Chen et al. 2020.


**Ethics Review Area:**

["I don’t know"]

**Limitations:**

I do not see much improvement from this method in comparison to previous methods. In fact in many settings the improvement is negligible such as in Table 1. Moreover, sometimes it is not practical to have access to the base class samples during fine-tuning stages and the assumption. Therefore source free fine-tuning is not practical with this method.

The pseudolabel creation for base class samples with a classifier initially built on novel target samples is cumbersome as the label set do not match in most cases. Therefore, the filtering step on top of this assumption with entropy criterion does not make sense with a reasonably big domain between base classes and novel classes. Also, many standard benchmark datasets are missing in the evaluation such as tieredImagenet, CUB, domain shift experiments such as mini-imagenet to CUB.

**Strengths And Weaknesses:**

The papers' idea of using manifold mix up is complemented by the use of hard sampling for further use during fine-tuning, which is simple and interesting.

---

> ### Author Response · Authors · 2022-08-02
> **Thank you for the valuable feedback. Comments are provided below.**
>
> **Q1. [... eq. 1 just shows cross entropy where is the full loss according to the description]:**
>
> Ans:  We make use of a two stage training process, following [9, 19]. The self-supervised loss is just used to pre-train the model using the approach [9]. The pre-trained backbone is then used for our approach which uses the cross-entropy loss.
>
> **Q2. [How exactly the entropy threshold for filtering pseudolabels for base classes is selected?]**
>
> Ans: Entropy-threshold is a hyperparameter for our approach. We observed that the entropy values for most cases were in the range [0, 1.65]. We then empirically set the value to 1.55, which seems to be the mode of the distribution of entropy values.
>
> **Q3. [ How this method is compared with the "Charting the Right Manifold: Manifold Mixup for Few-shot Learning" paper by Mangla et al. in WACV 2020?” ]**
>
> Ans: Our approach primarily addresses the data scarcity issue in FSL, while the mentioned paper (S2M2R [14]) uses manifold mixup as a regularizer across layers - without increasing the number of samples. Hence the motivation of using mixup is fundamentally different in the two works. Please refer to Page 3 (main paper), L107 for further discussion. Also, note that our method outperforms the best reported results in this paper for miniImageNet and CIFAR-FS datasets in both 5-way 5-shot and 5-way 1-shot setting as shown in the table below.
>
> **Table 3. Comparison between S2M2R and our method.**
>
> | Method     | miniImageNet | miniImageNet | CIFAR-FS   | CIFAR-FS   |
> |------------|--------------|--------------|------------|------------|
> |            | 1-shot       | 5-shot       | 1-shot     | 5-shot     |
> | S2M2R [14] | 64.93 $\pm$ 0.18   | 83.18 $\pm$ 0.11   | 74.81 $\pm$ 0.19 | 87.47 $\pm$ 0.13 |
> | FeLMi      | **67.47 $\pm$ 0.78**   | **86.08 $\pm$ 0.44**   | **78.22 $\pm$ 0.7**  | **89.47 $\pm$ 0.5**  |
>
>
>
> **Q4. [Why the other standard benchmarks such as tiered-ImageNet or CUB is missing”.]**
>
> Ans: Thanks for the suggestion. We have experimented with tieredImageNet and the results are shown above in Table 1.
> We see that our approach outperforms baseline in tieredimagenet in 5-way-5-shot setting. Due to the limited time and resources, instead of using CUB for few-shot learning, we used that dataset in the transfer learning setting as suggested and obtained significant improvements.
>
> **Table 1: Results on tieredImageNet (5-way 5-shot)**
>
> | Method      | Backbone  | 5-way 5shot |
> |-------------|-----------|-----------------|
> | ProtoNet [25]   | ResNet-12 | 83.40 $\pm$ 0.65    |
> | TapNet  [32]    | ResNet-12 | 80.26 $\pm$ 0.12    |
> | MetaOpt  [11]   | ResNet-12 | 81.56 $\pm$ 0.53    |
> | MTL     [26]    | ResNet-12 | 80.61 $\pm$ 0.90    |
> | Shotfree [17]   | ResNet-12 | 82.64 $\pm$ 0.43    |
> | DSN-MR   [24]   | ResNet-12 | 82.85 $\pm$ 0.56    |
> | DeepEMD   [34]  | ResNet-12 | 86.03 $\pm$ 0.58    |
> | FEAT    [31]    | ResNet-12 | 84.79 $\pm$ 0.16    |
> | RFS-Simple [28]  | ResNet-12 | 84.41 $\pm$ 0.55    |
> | RFS-distill [28]| ResNet-12 | 86.03 $\pm$ 0.49     |
> | AssoAlign [1]  | ResNet-18 | 85.97 $\pm$ 0.49    |
> | SKD-Gen1 [16]  | ResNet-12 | 86.61 $\pm$ 0.59    |
> | InfoPatch [8]  | ResNet-12 | 85.44 $\pm$ 0.35    |
> | IEPT     [36]   | ResNet-12 | 86.73 $\pm$ 0.34    |
> | IER-distill [19] | ResNet-12 | 86.57 $\pm$ 0.81    |
> | LabelHalluc [9] | ResNet-12 | 86.80 $\pm$ 0.58    |
> | FeLMi  (Ours)     | ResNet-12 | **87.07 $\pm$ 0.55**    |
>
>
>
> **Q5. [different architechtures such as WRN]**
>
> Ans:  We use Resnet-12, following several recent papers [9, 13, 19]. Further, as various recent works [9, 13, 19] have shown, Resnet-12 backbone outperforms the WRN baseline.
>
> The next part will be commented below

---

> > ### Author Response · Authors · 2022-08-02
> > **More comments**
> >
> > **Q6. [Also, are the methods compared are evaluated based on the same pretrained model?]**
> >
> > Ans: The recent methods [IER [19], LabHal [9]] make use of the same pre-trained models. We are also using the same pretrained model.
> >
> > **Q7. [Over, how many episodes the average accuracies are reported]**
> >
> > Ans: Thanks for pointing it out. The average accuracy has been computed over 600 episodes following the recent methods [9, 19]. We will update this in the final version.
> >
> > **Q8. [domain-shift results for example mini to CUB setting]**
> >
> > Ans: Thanks for this suggestion. We have now included results for the cross-domain transfer setting.
> >
> > **Table 2: Results for miniImageNet -> CUB**
> >
> > | Method     | Backbone  | 1-shot     | 5-shot     |
> > |------------|-----------|------------|------------|
> > | Baseline++ | ResNet-18 | 40.44 $\pm$ 0.75 | 56.64 $\pm$ 0.72 |
> > | MetaOPt [11]  | ResNet-18 | 44.79 $\pm$ 0.75 | 64.98 $\pm$ 0.68 |
> > | S2M2R [14]     | ResNet-18 | 48.24 $\pm$ 0.84 | 70.44 $\pm$ 0.75 |
> > | AssoAlign [1] | ResNet-18 | 47.25 $\pm$ 0.76 | 72.37 $\pm$ 0.89 |
> > | MixFSL     | ResNet-18 | -          | 68.77 $\pm$ 0.9  |
> > | MT-ConFT   | ResNet-10 | 49.25 $\pm$ 0.83 | 74.45 $\pm$ 0.71 |
> > | FelMi (Ours)    | ResNet-12 | **51.66 $\pm$ 0.82** | **77.61 $\pm$ 0.69** |
> >
> > **References:**
> >
> > [Baseline++] Chen et al. " A closer look at few-shot classification". International Conference on Learning Representations, 2019
> >
> > [MixFSL] Afrasiyabi et al. "Mixture-based feature space learning for few-shot Â image classification." International Conference on Computer Vision, 2021
> >
> > [MT-ConFT] Das et al. "On the Importance of Distractors for Few-Shot Classification ", International Conference on Computer Vision, 2021
> >
> >
> > We observe a significant gain in performance for the cross-domain settings.
> >
> > **Q9. [ I do not see much improvement from this method in comparison to previous methods.]**
> >
> > Ans:  While our results are competitive with other approaches in in-domain settings, our approach is **simple and effective**. Also, as evident from the cross-domain setting we outperform the prior approaches by a **large margin**. This shows the efficacy of our approach of using hard-negative mixup.
> >
> > **Q10. [Not practical to have access to the base class samples during fine-tuning stages and the assumption.]**
> >
> > Ans: The motivation of using base class examples in addition was to combat the inherent data scarcity problems addressed in [19]. We are also considering this setting.
> > Note that, this additional data need not to be from base class, the source could be anything. Therefore, we are utilizing additional unrelated data by pseudolabeling, which could potentially combine few-shot learning with challenging semi-supervised or unsupervised learning frameworks.
> >
> > **Q11. [Therefore, the filtering step on top of this assumption with entropy criterion does not make sense with a reasonably big domain between base classes and novel classes]**
> >
> > Ans:  We have observed that our approach performs better in the cross-domain setting (miniImageNet -> CUB). However, we do understand that psuedolabeling based methods [9] would fail when dealing with extremely large domain shift between base and novel. We will update the limitations in the paper to reflect this. Thanks for the suggestion.

---

### Official Review · Reviewer_7gPu · 2022-07-12

**Rating:** 8
**Confidence:** 5
**Soundness:** 3 good
**Presentation:** 4 excellent
**Contribution:** 3 good

**Summary:**

This work presents a study on applying 1) combining pseudo-labeling with entropy-based label filtering for representation learning and 2) novel-novel and base-novel manifold mixup with entropy-based filtering for adapting base representation to novel classes for improving few-shot image recognition tasks. Under appropriate hyperparameter settings, the proposed approach achieves competitive performance on standard few-shot image recognition benchmarks. Ablation studies are conducted to investigate the gains brought by individual techniques.

**Questions:**

See weaknesses.

**Limitations:**

Limitations are addressed adequately.

**Strengths And Weaknesses:**

Strengths:

1) The paper is well-written and easy to follow. Approaches, experiment settings and implementation details are clearly described, in a way that helps reproducibility of the proposed work. Experiment results are well organized.
2) Experiments and ablation studies seem thorough. Standard benchmarks are used. Latest works that follow the same experimental settings are included as baselines. Necessary baselines are included. Ablation studies included all components of the system.
3) Experiment results seem to show highly competitive performance (without transductive learning).

Weaknesses:

1) While it is clear that the proposed approach worked, it is not very clear how it worked. Here are my recommendations
- For entropy-based filtering, try to fit the qualitative examples in the main paper
- Show some hard examples selected by mixup. Maybe something like (0.9x image A + 0.1x image B)
- Which classes often benefit from pseudo-labeing, mixup and entropy filtering?

2) While not necessary, it would be interesting to learn about the sensitivity of the proposed approach to hyperparameters. The use of different hyperparameters for 1- and 5-shot might be a deviation from classical setups? Although it's not entirely surprising that different hyperparameters might be necessary as the dataset size changes. It would help if measures are taken to control overfitting to specific benchmarks.


=============================

Author response adequately addresses my concerns.

---

> ### Author Response · Authors · 2022-08-02
> **Thank you for the positive feedback.**
>
> **Q1. [ try to fit the qualitative examples in the main paper]**
>
> Ans: Thanks for the suggestion. We will update the paper using the qualitative examples in the final version.
>
>
> **Q2. [Show some hard examples selected by mixup]**
>
> Ans: It is to be noted that we mix examples in the feature space rather than at the input. For this reason, it is difficult to trace back the generated mixed up feature to the corresponding image. Instead, we visualize the feature space with novel examples, base examples and mixed up examples through tSNE plots. Please refer to Supplementary Page 7, Fig 3. We see that our approach generates examples which are close to the boundary between novel classes thus helping with the learning process.
>
>
> **Q3. [Which classes often benefit from pseudo-labeling, mixup and entropy filtering]**
>
> Ans: Thanks for the suggestion. This analysis would help us to comprehend the efficacy of our proposed approach.
>
> o	**Pseudo-labeling:** The most informative pseudo-labels are obtained when the base class samples are semantically close to the novel classes. For example in the case of miniImageNet, base examples class "Arctic Fox" has been assigned as closest confident pseudolabels for novel class "Malamute", which are both dog classes, therefore semantically closer (refer to Fig. 4 in the supplementary material). Hence, using these semantically closer samples would increase the performance of the Few-shot learner.
>
> o	**Entropy filtering:** Using base examples might be helpful, but unrelated base examples could potentially harm the performance as well. Through this approach we filter out base examples with high entropy - i.e., base samples with large distribution gap would be removed.
>
> For example, consider a 5-way classification episode with classes, e.g., "crate", "vase", "lion", "african hunting dog" and "malamute" (refer Fig. 4 in the supplementary material). Any class in the base examples which is close to any of the novel classes would be helpful for this 5-way classification, but base classes totally unrelated to any of the novel classes would be confusing for the few-shot learner. For example "Artic Fox" class pseudolabeled as "malamute" would be helpful. However, "toucan", "robin" (bird species) which are not related to any of these classes would be confusing and produces high entropy. Therefore, entropy filtering would remove such classes to improve the performance.
>
>
> o	**Mixup:** Mixup provides robustness in classification. Note, here we are using both novel-novel and base-novel mixup, to generate new samples close to the novel distribution. For base-novel mixup, this is ensured by weighting the novel samples more than the base examples during mixup generation.
> Intuitively, base-novel mixup helps generate more samples close to the novel distribution but with some attributes from the base set.
> Moreover, we use simple margin based hard-mixup to select samples which are closer to the decision boundary and hence helps to widen the decision boundary and the classification as well.
> In the tSNE provided in the Fig. 3 of the supplementary, we observe that the generated hard samples concentrate near the decision boundary in the case of 5-way classification.
>
>
> **Q4. [Sensitivity of the proposed approach to hyperparameters]**
>
> Ans: Please refer to Fig. 2 and Fig. 3 in the main paper and Fig. 1 and Fig. 2 in the supplementary material for sensitivity of the results to the most important hyperparameters (beta distribution parameter (alpha) and number of mixup samples (N)).
>
> **Q5. [While not necessary, it would be interesting to learn about the sensitivity of the proposed approach to hyperparameters. The use of different hyperparameters for 1- and 5-shot might be a deviation from classical setups? Although it's not entirely surprising that different hyperparameters might be necessary as the dataset size changes. It would help if measures are taken to control overfitting to specific benchmarks.]**
>
>
> Ans: We agree with the reviewer and follow the standard setup (ref [9], [19] in the main paper) of searching for optimal hyperparameters for both the 5-way 5-shot and 5-way 1-shot settings separately.

---

> > ### Comment · Reviewer_7gPu · 2022-08-09
> > **Response to author comments**
> >
> > Thanks for the reply. The response adequately addressed my concerns.

---

> > > ### Author Response · Authors · 2022-08-09
> > > **Response to reviewer 7gPu**
> > >
> > > Thanks for the insightful comments and positive feeback. We would also like to point out that we have performed additional experiments on large-scale dataset and cross-domain few shot benchmarks and demonstrate state-of-the-art results in Table 1 and Table 2 respectively. We have validated our method with statistical significance test as well.
> > >
> > >
> > > **Table 1: Results on tieredImageNet (5-way 5-shot)**
> > >
> > > | Method      | Backbone  | 5-way 5shot |
> > > |-------------|-----------|-----------------|
> > > | ProtoNet [25]   | ResNet-12 | 83.40 $\pm$ 0.65    |
> > > | TapNet  [32]    | ResNet-12 | 80.26 $\pm$ 0.12    |
> > > | MetaOpt  [11]   | ResNet-12 | 81.56 $\pm$ 0.53    |
> > > | MTL     [26]    | ResNet-12 | 80.61 $\pm$ 0.90    |
> > > | Shotfree [17]   | ResNet-12 | 82.64 $\pm$ 0.43    |
> > > | DSN-MR   [24]   | ResNet-12 | 82.85 $\pm$ 0.56    |
> > > | DeepEMD   [34]  | ResNet-12 | 86.03 $\pm$ 0.58    |
> > > | FEAT    [31]    | ResNet-12 | 84.79 $\pm$ 0.16    |
> > > | RFS-Simple [28]  | ResNet-12 | 84.41 $\pm$ 0.55    |
> > > | RFS-distill [28]| ResNet-12 | 86.03 $\pm$ 0.49     |
> > > | AssoAlign [1]  | ResNet-18 | 85.97 $\pm$ 0.49    |
> > > | SKD-Gen1 [16]  | ResNet-12 | 86.61 $\pm$ 0.59    |
> > > | InfoPatch [8]  | ResNet-12 | 85.44 $\pm$ 0.35    |
> > > | IEPT     [36]   | ResNet-12 | 86.73 $\pm$ 0.34    |
> > > | IER-distill [19] | ResNet-12 | 86.57 $\pm$ 0.81    |
> > > | LabelHalluc [9] | ResNet-12 | 86.80 $\pm$ 0.58    |
> > > | FeLMi (Ours)      | ResNet-12 | **87.07 $\pm$ 0.55**    |
> > >
> > >
> > >
> > > **Table 2: Results for Cross-domain Few-shot Learning ( miniImageNet -> CUB)**
> > >
> > > | Method     | Backbone  | 1-shot     | 5-shot     |
> > > |------------|-----------|------------|------------|
> > > | Baseline++ | ResNet-18 | 40.44 $\pm$ 0.75 | 56.64 $\pm$ 0.72 |
> > > | MetaOPt [11]  | ResNet-18 | 44.79 $\pm$ 0.75 | 64.98 $\pm$ 0.68 |
> > > | S2M2R [14]     | ResNet-18 | 48.24 $\pm$ 0.84 | 70.44 $\pm$ 0.75 |
> > > | AssoAlign [1] | ResNet-18 | 47.25 $\pm$ 0.76 | 72.37 $\pm$ 0.89 |
> > > | MixFSL     | ResNet-18 | -          | 68.77 $\pm$ 0.9  |
> > > | MT-ConFT   | ResNet-10 | 49.25 $\pm$ 0.83 | 74.45 $\pm$ 0.71 |
> > > | FelMi (Ours)    | ResNet-12 | **51.66 $\pm$ 0.82** | **77.61 $\pm$ 0.69** |
> > >
> > >
> > > **References:**
> > >
> > > [Baseline++] Chen et al. " A closer look at few-shot classification". International Conference on Learning Representations, 2019
> > >
> > > [MixFSL] Afrasiyabi et al. "Mixture-based feature space learning for few-shot Â image classification." International Conference on Computer Vision, 2021
> > >
> > > [MT-ConFT] Das et al. "On the Importance of Distractors for Few-Shot Classification ", International Conference on Computer Vision, 2021

---

### Author Response · Authors · 2022-08-02
**Thanks for the feedback. Please see the comments.**

We thank all the esteemed reviewers for their insightful comments.

**Positive points:**

**1. Simple and effective:** [7gPu], [3CtS], [2Qz6], [WHpk]

**2. Well-written and easy to follow:** [7gPu],[2Qz6], [WHpk]

**3. Strong Experimental results:** [7gPu], [WHpk]

Also, based on the reviewers' recommendation, we have performed experiments on larger dataset, e.g., tieredImageNet and also verified the efficacy of our approach in cross-domain settings (e.g., miniImageNet -> CUB). We have obtained consistent improvements in these experiments and provide some insights of our method.

**Exp 1. Results on tieredImagenet**

As suggested, we experimented on the large scale benchmark -- tiered-ImageNet, and demonstrate state of the art performances.

**Table 1: Results on tieredImageNet (5-way 5-shot)**

| Method      | Backbone  | 5-way 5shot |
|-------------|-----------|-----------------|
| ProtoNet [25]   | ResNet-12 | 83.40 $\pm$ 0.65    |
| TapNet  [32]    | ResNet-12 | 80.26 $\pm$ 0.12    |
| MetaOpt  [11]   | ResNet-12 | 81.56 $\pm$ 0.53    |
| MTL     [26]    | ResNet-12 | 80.61 $\pm$ 0.90    |
| Shotfree [17]   | ResNet-12 | 82.64 $\pm$ 0.43    |
| DSN-MR   [24]   | ResNet-12 | 82.85 $\pm$ 0.56    |
| DeepEMD   [34]  | ResNet-12 | 86.03 $\pm$ 0.58    |
| FEAT    [31]    | ResNet-12 | 84.79 $\pm$ 0.16    |
| RFS-Simple [28]  | ResNet-12 | 84.41 $\pm$ 0.55    |
| RFS-distill [28]| ResNet-12 | 86.03 $\pm$ 0.49     |
| AssoAlign [1]  | ResNet-18 | 85.97 $\pm$ 0.49    |
| SKD-Gen1 [16]  | ResNet-12 | 86.61 $\pm$ 0.59    |
| InfoPatch [8]  | ResNet-12 | 85.44 $\pm$ 0.35    |
| IEPT     [36]   | ResNet-12 | 86.73 $\pm$ 0.34    |
| IER-distill [19] | ResNet-12 | 86.57 $\pm$ 0.81    |
| LabelHalluc [9] | ResNet-12 | 86.80 $\pm$ 0.58    |
| FeLMi  (Ours)     | ResNet-12 | **87.07 $\pm$ 0.55**    |




**Exp 2. Results on cross-domain FSL (miniImageNet -> CUB)**

We have also explored the cross-domain few-shot setting and demonstrate state-of-the-art results.

**Table 2: Results for miniImageNet -> CUB**

| Method     | Backbone  | 1-shot     | 5-shot     |
|------------|-----------|------------|------------|
| Baseline++ | ResNet-18 | 40.44 $\pm$ 0.75 | 56.64 $\pm$ 0.72 |
| MetaOPt [11]  | ResNet-18 | 44.79 $\pm$ 0.75 | 64.98 $\pm$ 0.68 |
| S2M2R [14]     | ResNet-18 | 48.24 $\pm$ 0.84 | 70.44 $\pm$ 0.75 |
| AssoAlign [1] | ResNet-18 | 47.25 $\pm$ 0.76 | 72.37 $\pm$ 0.89 |
| MixFSL     | ResNet-18 | -          | 68.77 $\pm$ 0.9  |
| MT-ConFT   | ResNet-10 | 49.25 $\pm$ 0.83 | 74.45 $\pm$ 0.71 |
| FelMi (Ours)    | ResNet-12 | **51.66 $\pm$ 0.82** | **77.61 $\pm$ 0.69** |


**References:**

[Baseline++] Chen et al. " A closer look at few-shot classification". International Conference on Learning Representations, 2019

[MixFSL] Afrasiyabi et al. "Mixture-based feature space learning for few-shot Â image classification." International Conference on Computer Vision, 2021

[MT-ConFT] Das et al. "On the Importance of Distractors for Few-Shot Classification ", International Conference on Computer Vision, 2021

---

### Meta-Review · Area_Chair_QX1q · 2022-08-25

**Recommendation:** Accept
**Confidence:** Certain

**Metareview:**

The submission introduces an approach to few-shot learning called Few-Shot Learning with Hard Mixup (FeLMi) which, as its name suggests, applies hard manifold mixup as an augmentation strategy for adapting a pre-trained model to a small training set of downstream examples. The model is first trained on the base classes using a combination of supervised learning and Invariant and Equivariant Representation learning (IER), then a linear classifier is trained on top of the frozen backbone using the novel classes' support set and pseudolabels are generated for the entire base class dataset. Base class examples are filtered to exclude ones with low pseudolabel entropy (using a thresholding hyperparameter). Feature-level mixup is applied on base-novel and novel-novel example pairs, and the resulting examples are subsampled to the N hardest ones based on the difference in top-2 probabilities. The model is then fine-tuned on the pseudolabeled base examples, novel examples, and hard-mixup examples.

Results are presented on two CIFAR100-based few-shot classification benchmarks (CIFAR-FS, FC-100) and mini-ImageNet in the 5-way, 1-shot and 5-way, 5-shot settings. FeLMi is shown to outperform competing approaches. Ablation analyses are also presented to assess the contribution of various components on performance improvements.

Reviewers highlight the submission's writing quality and clarity (7gPu, 2Qz6, WHpk). Opinions are split on how straightforward the proposed approach is, with Reviewers 3CtS and WHpk noting its simplicity, and Reviewer 2Qz6 expressing concerns over its many moving parts. Opinions are also split on the significance of the performance improvements; Reviewer 7gPu finds FeLMi's performance competitive with competing approaches, and Reviewers 3CtS and WHpk are concerned that the improvements are modest. The authors respond by emphasizing that FeLMi is simple and effective, but Reviewer 3CtS remains eager to see a clearer performance gap. Reviewer 3CtS is also concerned that the approach is not source-free, to which the authors respond that the unlabeled data could also come from another source than the upstream training dataset.

Following the discussions, opinions remain divided among reviewers, although the majority is either leaning towards or strongly recommending acceptance. Reviewer 3CtS still recommends rejection, but is open to an acceptance recommendation. I therefore recommend acceptance.

**Award:**

No

---

### Decision · Program_Chairs · 2022-09-14

Accept